# Sustainable land management enhances ecological and economic multifunctionality under ambient and future climate

Friedrich Scherzinger [1], Martin Schädler [1,2], Thomas Reitz [1,3], Rui Yin [1,2,4], Harald Auge [1,2], Ines Merbach[2], Christiane Roscher [1,5], W Stanley Harpole [1,5,6], Evgenia Blagodatskaya[3], Julia Siebert[1], Marcel Ciobanu[7], Fabian Marder [1], Nico Eisenhauer [1,4,9] ✉ & Martin Quaas [1,8,9]

The currently dominant types of land management are threatening the multifunctionality of ecosystems, which is vital for human well-being. Here, we present a novel ecological-economic assessment of how multifunctionality of agroecosystems in Central Germany depends on land-use type and climate. Our analysis includes 14 ecosystem variables in a large-scale field experiment with five different land-use types under two different climate scenarios (ambient and future climate). We consider ecological multifunctionality measures using averaging approaches with different weights, reflecting preferences of four relevant stakeholders based on adapted survey data. Additionally, we propose an economic multifunctionality measure based on the aggregate economic value of ecosystem services. Results show that intensive management and future climate decrease ecological multifunctionality for most scenarios in both grassland and cropland. Only under a weighting based on farmers' preferences, intensively-managed grassland shows higher multifunctionality than sustainably-managed grassland. The economic multifunctionality measure is about ~1.7 to 1.9 times higher for sustainable, compared to intensive, management for both grassland and cropland. Soil biodiversity correlates positively with ecological multifunctionality and is expected to be one of its drivers. As the currently prevailing land management provides high multifunctionality for farmers, but not for society at large, we suggest to promote and economically incentivise sustainable land management that enhances both ecological and economic multifunctionality, also under future climatic conditions.

Land use and climate change are major drivers of ecosystem functioning and the provision of ecosystem services, which are vital for human well-being. Ecosystem functions are natural processes (biological and geochemical) that are indirectly linked to the provision of ecosystem services and the economic value they generate for ecosystem managers and society at large[1]. Ecosystem multifunctionality describes the ability of ecosystems to provide multiple functions and services simultaneously[2]. Due to the interdependent and overlapping

---

nature of different functions, multifunctionality is more than just the sum of its parts[3]. The field of multifunctionality research has been gaining more and more attention[3,4], and increasingly the trade-offs and synergies that occur between different ecosystem functions and services are being recognised[4,5]. The multifunctionality perspective also allows for a more comprehensive assessment of the impact of global change on ecosystems and on the benefits that stakeholders obtain from the ecosystem.

Ecological multifunctionality measures commonly use averaging approaches, with weighting schemes reflecting stakeholders' preferences for different functions and associated ecosystem services[6]. Here, we additionally propose an economic multifunctionality measure that is based on the economic value of services to society at large. This includes the farmer, who benefits from yield, and many others who benefit from services such as climate regulation, carbon sequestration, and maintenance of water quality. Here, we study how land management and climate change affect ecosystem multifunctionality, contrasting alternative ecological multifunctionality measures and the economic multifunctionality measure. The consideration of both ecological and economic multifunctionality indices overcomes the shortcomings of a purely economic evaluation of ecosystems. While an economic evaluation allows an assessment of the benefits for society at large with a unified, monetary metric, it often fails to capture the importance of different aspects of ecosystems, e.g., of cultural ecosystem services[7,8]. Conversely, ecological multifunctionality measures can assess ecosystem services based on their relative importance to different stakeholder groups, which allows consideration of other aspects such as overall community-level multifunctionality[9,10]. Other than traditional (non-weighted) ecological multifunctionality measures, both the ecological multifunctionality approach and the economic multifunctionality approach (by differentiating benefits for farmers and for society at large) allow assessing how well an ecosystem meets the priorities and demands of different stakeholders.

Previous research has identified land use and anthropogenic climate change as drivers of changing ecosystem multifunctionality[4]. Climate change is expected to have varying, net negative effects on ecosystem services[11,12]. As different ecosystem services are valued differently by stakeholders, the overall impact of climate change on stakeholders and economic benefit of society at large remains unclear[4]. Land-use intensification (increased application of agrochemicals and machinery) has been found to decrease multifunctionality[6,13–15]. In croplands, organic farming is expected to increase multifunctionality compared to conventional farming, as it promotes regulating and supporting services instead of only a small number of provision services[16].

Besides climate and land-use change, biodiversity loss was identified as a major driver of declining multifunctionality of agricultural ecosystems[6,14]. Biodiversity increases and stabilises many different natural processes, including ecosystem functions, and ecosystem services associated with them[17–20]. However, the role of biodiversity is ambiguous, as it is both a driver of many different ecosystem functions and a function (or service) by itself[21]. While biodiversity itself is negatively affected by land-use and climate change[22], also the strength of the biodiversity-multifunctionality relationship may be dependent on changes in environmental conditions[23]. However, empirical work on this topic is scarce, and a study that simulated future climate with elevated $CO_2$ concentration and enhanced nitrogen deposition found no significant difference in the magnitude of the effect of biodiversity on soil multifunctionality under ambient compared to future climate conditions[19]. Contrary, a stronger effect of biodiversity on multifunctionality was observed under more stressful environments caused by global change drivers (indicating future climate conditions), implying ecosystems with higher biodiversity are indeed more resistant to future climate[18,24,25]. Therefore, our study aims to determine how the biodiversity-multifunctionality relationship changes under future climate across important land-use types in Central Europe.

Most studies in the context of multifunctionality research have focused on single drivers of multifunctionality (e.g., the effect of microbial diversity on (soil) multifunctionality[26,27], the effect of biodiversity on multifunctionality[28,29], or the change of (soil) multifunctionality across a natural climate gradient[4,30]). Still, no study has investigated the combined effect of different land-use types and future climate on multifunctionality in the context of agriculture by making use of mixed ecological-economic multifunctionality measures that consider varying stakeholder preferences for different ecosystem services combined with a direct quantification of the ecosystem service provision for society at large in monetary units. Here, we address this research gap in order to provide a more comprehensive understanding of the complex interplay between land-use and climate change, biodiversity, and ecosystem multifunctionality in agricultural ecosystems, which may ultimately guide informed decision-making for land management. See Table 1 for an overview of key terms.

We analyse data for the years 2014–2023 from the Global Change Experimental Facility (GCEF) in Saxony-Anhalt, Germany. The GCEF is a large field experiment with orthogonal manipulation of climate (ambient

**Table 1 | Glossary of key terms**

| | |
|---|---|
| Ecosystem function | Natural processes (biological and geochemical) that can be used as indicators for the provision of ecosystem services. |
| Ecosystem service | The benefits (goods and services) people obtain from ecosystems. |
| Ecosystem multifunctionality | The ability of ecosystems to provide multiple functions or services simultaneously. |
| Ecological ecosystem multifunctionality | Multifunctionality is calculated as the weighted average of the normalized levels of different ecosystem functions. Here, weightings are based on measured preferences of four stakeholder groups (farmers, local residents, environmental conservation agencies, tourism), derived from survey data[41] that were adapted to the ecosystem services assessed in this study. To ensure consistency with previous studies, we added a scenario of equal weighting of each ecosystem function and a scenario of equal weighting of each ecosystem service (the latter represents a weighting of each ecosystem function according to their relative share of the respective ecosystem service (e.g., as each of the six ecosystem services is weighted with 1/6 the three functions that make up the service 'soil biodiversity conservation' are weighted with 1/18 each, see Fig. 2 and Methods). Visualised in shades of blue. |
| Economic ecosystem multifunctionality | Multifunctionality calculated as the sum of the monetary values of the ecosystem services assessed. The ecosystem service 'landscape aesthetics' was not monetised due to lack of data availability. Visualised in shades of orange. |
| Economic ecosystem multifunctionality (farmers) | The monetary ecosystem service value farmers directly receive from the ecosystem (consisting of food production and the two insurance-providing ecosystem services biodiversity and soil health that have an economic value (willingness to pay) for the risk-averse farmer on top of the productivity effects). Visualised in shades of orange in Fig. 2. |
| Sustainable management | Land management refraining from the application of mineral nitrogen fertiliser and pesticides. |
| Intensive management | Land management making use of the application of mineral nitrogen fertiliser and pesticides. |

and future climate) and five different land-use types on 50 plots of approximately 400 m² each[31]. The experiment includes three grassland and two cropland types of land use, with sustainable management (without application of mineral nitrogen fertiliser and pesticides) and intensive management (with application of mineral nitrogen fertiliser and

| | Sustainable management | Intensive management |
|---|---|---|
| **Grassland** | Extensive meadow (EM) | Intensive meadow (IM) |
| | Extensive pasture (EP) | |
| **Farmland** | Organic farming (OF) | Conventional farming (CF) |

**Fig. 1 | Overview of the land-use types considered in this study.** Classification into sustainable and intensive management was done based on the input of agro-chemicals (mineral nitrogen fertiliser and pesticides) which was refrained under sustainable management.

pesticides), see Fig. 1. The 50 plots are grouped in 10 mainplots, whereas each mainplot contains all five land use types, and five mainplots are managed under ambient climate and five mainplots under future climate (i.e., 5 replications for each combination of land use and climate type).

We measure 14 ecosystem functions corresponding to six ecosystem services (Fig. 2). For simplicity we will refer to these variables as "ecosystem functions", but we acknowledge that some of them are biodiversity metrics and not ecosystem functions sensu stricto. The classification and categorisation of ecosystem functions is based on the Millennium Ecosystem Assessment Program[32]. Aboveground biomass production ('yield') indicates food production; total organic soil carbon (yearly flux) indicates climate regulation; soil nutrient concentration indicates water quality; microbial biomass, enzymatic activity, and decomposition rate indicate soil health, as they enhance the nutrient availability for plants and the soil water retention[19,33], see Methods for details. The biodiversity of soil nematodes and soil meso- /

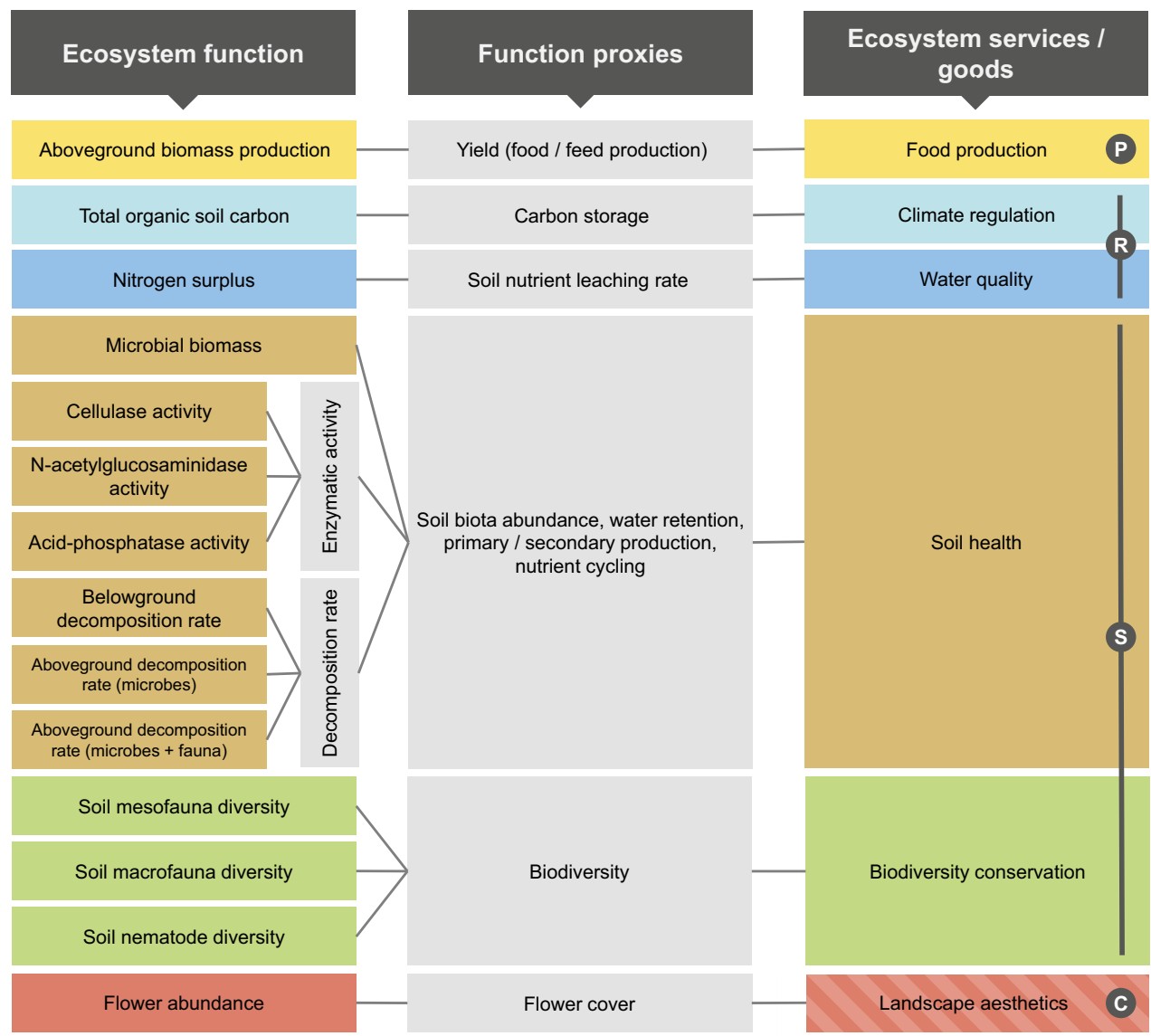

**Fig. 2 | Conceptual framework for the calculation of ecosystem multi-functionality based on 14 ecosystem variables approximating six ecosystem services / goods (P: provisioning service, R: regulating services, S: supporting services, C: cultural service) with both an ecological and an economic approach.** For the ecological approach of multifunctionality calculation, ecosystem variable levels are used to calculate ecosystem service levels (according to the ecosystem functions' share of the respective service they are approximating). For the economic approach, the monetary value of the ecosystem services is used. Flower abundance was integrated into the ecological multifunctionality index, but its corresponding ecosystem service 'landscape aesthetics' was not integrated into the economic multifunctionality index (hatched rectangle).

macrofauna indicate biodiversity conservation. In the context of our study, biodiversity conservation serves a dual purpose in ecosystem service valuation. Firstly, as biodiversity is found to reduce variability of aboveground biomass production, biodiversity provides a natural insurance for farmers[34–36]. We quantify the natural insurance value of biodiversity and include it as part of the economic multifunctionality calculation. Secondly, biodiversity has an intrinsic value[21]. Our focus is on biodiversity of soil organisms. Although these are not typically the primary organisms that people value, people do appreciate soil biodiversity for its intrinsic value[37]. Due to a lack of data for this study region, a transfer of the intrinsic value of biodiversity into monetary units was not possible and we could not include an intrinsic value of biodiversity in the economic multifunctionality measure. In contrast, the ecological multifunctionality index includes the overall level of soil biodiversity, thus also capturing the intrinsic value of biodiversity. By that, the use of two distinct multifunctionality measures allowed us to evaluate different facets of biodiversity's value[21,38]. Flower abundance indicates landscape aesthetics, as people prefer flowering stages of landscapes[39]. An increase in the level of the ecosystem functions is desirable (except for nitrogen surplus), as it indicates an increase in the respective ecosystem services.

To measure ecological multifunctionality, we used the averaging approach[3]. For every ecosystem function, we calculated mean values over multiple years (Table S1), in line with previous ecosystem multifunctionality assessments[40]. We calculated the ecological multifunctionality on the plot level using the weighted average of the normalised levels of the 14 ecosystem functions. The weighting is based on stated preferences of four different stakeholder groups. We adapted survey data[41] (Fig. S1) to align the weighting factors with the set of ecosystem services assessed in this study. Within the adapted weighting factors, farmers and local residents put a higher weight on the provisioning service (food production), local residents and the tourism sector put a higher weight on the cultural service (landscape aesthetics), and environmental conservation agencies put a higher weight on supporting and regulating services. Further, we use both an equal weighting scenario of all 14 ecosystem functions, and an equal weighting scenario of all six ecosystem services to ensure comparability with previous studies[6] (Fig. S2; Table S2). For the calculation of economic multifunctionality, we monetised the ecosystem services listed in Fig. 2 (with the exception of landscape aesthetics, where data availability did not allow monetisation). We tested the sensitivity of the results of the economic multifunctionality index towards alternative price scenarios / social cost scenarios of yield, and $CO_2$ and nitrogen emission (Fig. S3). Further, we tested if soil biodiversity is related to ecological multifunctionality, as shown in previous studies that demonstrate the need of land managers to conserve soil biodiversity[18,19,26,27,29], and we tested if the strength of the relationship between biodiversity and ecological multifunctionality changes under climate change using a linear regression model. Our results may provide decision-making support to implement strategies to control and counteract the effects of land-use intensification and climate change on multifunctionality, to adapt land-use composition to provide high multifunctionality under a changing climate, and to steer policy decisions accordingly.

## Results
### Effects of climate and land-use type on ecological multifunctionality measures
Ecosystem functions show different responses to the land-use and climate treatments (Table S3; Fig. S4; Supplementary material, Section 8.1). The effect sizes of the different climate and land-use types on ecological ecosystem multifunctionality strongly depend on the weighting schemes of the four stakeholder groups considered in this study (Table S4, S5, and Supplementary Data Files 1–3, Fig. 3).

We begin by contrasting the land-use types grassland and cropland. Within this study, cropland has higher ecological multifunctionality than grassland. For three out of four weighting scenarios (farmers, local residents, tourism sector), under ambient climate, sustainably-managed cropland (organic farming) has the highest ecological multifunctionality of all land-use types. This is due to the high performance in the ecosystem services 'food production' and 'aesthetic value'. For the weighting scenario of environmental conservation agencies, sustainably-managed cropland and grassland show the same level of ecological multifunctionality. This indicates that management intensity has to be considered when assessing the effects of land-use types on ecological multifunctionality.

Overall, sustainable management increases ecological multifunctionality, compared to intensive management due to lower water quality under intensive management based on application of mineral fertiliser. The positive effect of sustainable management on ecological multifunctionality is observed for almost all weighting scenarios. Only for local residents, ecological multifunctionality for grasslands does not depend on management intensity, and only for farmers, intensive management increases ecological multifunctionality in grasslands, due to a higher economic yield.

Third, we contrast ambient and future climate. The main result is that ecological multifunctionality decreases under future climate. This is due to reduced water availability under future climate conditions that negatively affects major ecosystem functions, namely 'aboveground biomass production', 'flower abundance', and 'aboveground decomposition rate' (Fig. S4). Sustainably-managed cropland, despite showing the most pronounced absolute reduction of ecological multifunctionality under future climate, still shows the highest levels of ecological multifunctionality under future climate for three out of four weighting scenarios, highlighting its overall benefit under both ambient and future climate conditions.

Finally, we observe significant interaction effects between land-use type and climate on ecological multifunctionality for three out of four weighting scenarios (according to farmers', local residents', and tourism sector's preferences). For those three weighting scenarios, the detrimental effect of future climate on ecological multifunctionality is significantly more pronounced for sustainably-managed cropland compared to other land use types. The ecosystem function 'flower abundance' is strongly decreased under future climate for sustainably-managed cropland (organic farming), an effect that is negligible for the other land-use types, where flower abundance is low in general (Fig. S4; Table S3). Thus, flower abundance could be a key factor driving the interaction effect between climate and land-use type on ecological multifunctionality.

### Effects of climate and land-use type on economic multifunctionality measures
Contrary to the ecological measure of multifunctionality, future climate does not show any significant effect on economic multifunctionality (also under alternative price scenarios tested in the sensitivity analysis). However, the land-use and management type has a strong impact also on economic multifunctionality ($p < 0.001$***; Fig. 4). For both grassland ($p < 0.01$**) and cropland ($p < 0.001$***), economic multifunctionality is around -1.7 to 1.9 times higher for sustainable compared to intensive management due to the absence of mineral fertiliser application under sustainable management and thus a higher value of water quality for society at large. Contrary to ecological multifunctionality, economic multifunctionality is, on average, higher for grassland compared to cropland due to the higher carbon sequestration in grasslands resulting in a high economic value for the ecosystem service 'climate regulation'. The economic multifunctionality for farmers is significantly decreased under future

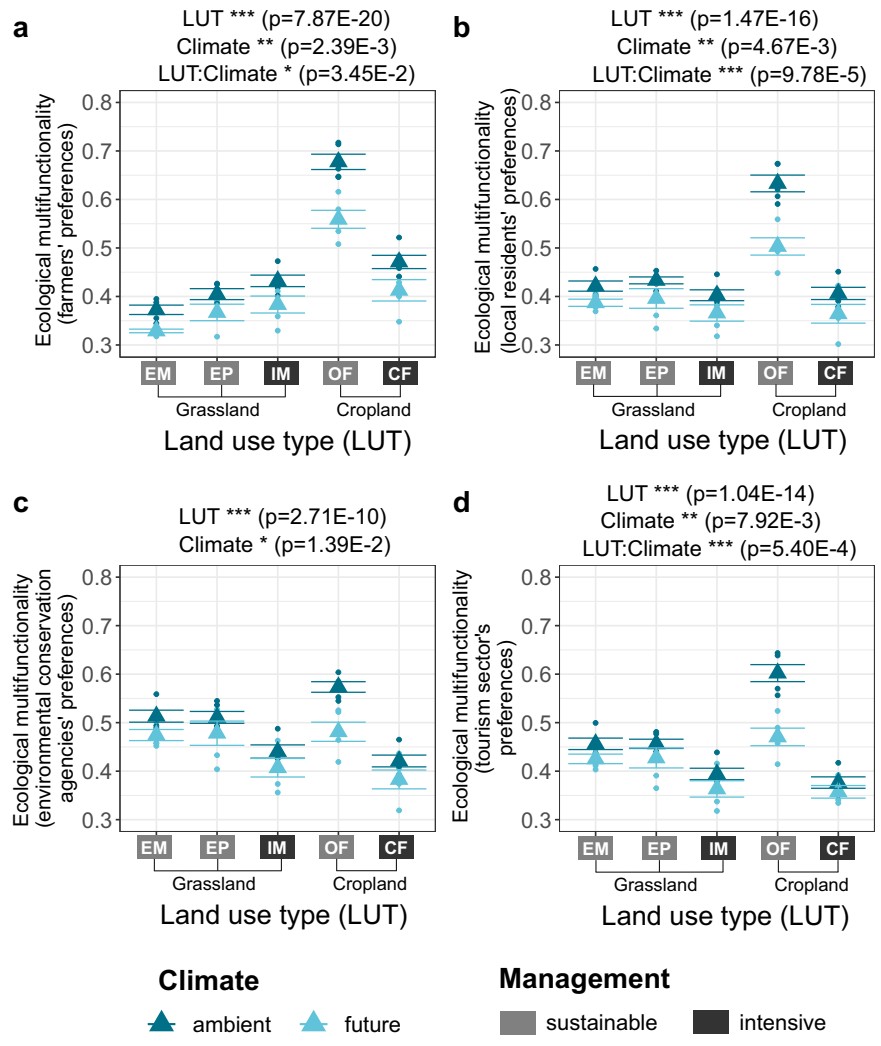

**Fig. 3 | Ecological multifunctionality under different land-use types.** Ecological ecosystem multifunctionality as affected by five different land-use types (EM: extensive meadow, EP: extensive pasture, IM: intensive meadow, OF: organic farming, CF: conventional farming) for both the ambient and the future climate and for four different weighting scenarios (**a** weighting according to farmers' preferences (highest weight on the provisioning service), **b** weighting according to local residents' preferences (higher weight on the provisioning and cultural services), **c** weighting according to environmental conservation agencies' preferences (higher weight on the regulating and supporting services), **d** weighting according to tourism sector's preferences (highest weight on the cultural service)). Dots indicate the multifunctionality level within the plots of the experiment (5 replicates for each LUT-Climate combination), triangles indicate the mean value of the respective LUT-Climate combination group). For statistical testing, F tests based on ANOVA (two-sided) without adjustments for multiple comparisons were used (numerator df: 4; denominator df: 32). Asterisks indicate a significant effect of the respective factor or interaction (*** $p < 0.001$; ** $p < 0.01$; * $p < 0.05$). Error bars represent the standard errors of the mean.

climate for all land use types (Table S6, Fig. 4), as high weight is assigned to the ecosystem function 'aboveground biomass', which is substantially negatively affected by future climate.

For sustainably-managed grassland, economic multifunctionality is around twice as high as the economic benefit for farmers, which includes only food production and the insurance value of biodiversity and soil health, but not the benefits of water quality and climate regulation that accrue to society at large. Under intensive management, by contrast, the economic benefit of farmers is higher than the economic benefit for society at large. The ecosystem service value composition within the economic multifunctionality measure differs between grassland and cropland (Fig. 5). For grassland, a significant proportion (28 to 70%) of economic multifunctionality is contributed by the service 'climate regulation', due to the higher carbon sequestration in grasslands. Under intensive management, the economic value of the service 'food production' is slightly increased for grassland, and slightly decreased for cropland. For the intensively-managed land-use types, nitrogen surplus – decreasing the value of water quality

– has a strongly negative impact on economic multifunctionality, an effect that is negligible for the sustainably-managed types. The monetary value of soil health depends on the level of biodiversity in the respective plot, and vice versa, due to the mutual effect of soil health and biodiversity on yield stability and its associated natural insurance value.

The finding that economic multifunctionality is higher for sustainable, compared to intensive, management is robust to different accounting prices for $CO_2$ emissions and to a 40% increase in crop prices. With the alternative accounting price for nitrogen leaching, the effect of higher economic multifunctionality for sustainable management under grassland and cropland types vanishes, while the effect of higher economic multifunctionality under grassland remains (Fig. S3, Table S9). This highlights the importance of site-specific characteristics when calculating social cost of nitrogen emissions.

Economic multifunctionality correlates with ecological multifunctionality under all weighting scenarios, including equal ecosystem

function and equal ecosystem service weighting. Correlation coefficients range from 0.352 (economic multifunctionality – local residents' ecological multifunctionality) to 0.738 (economic multifunctionality – environmental conservation agencies' ecological multifunctionality).

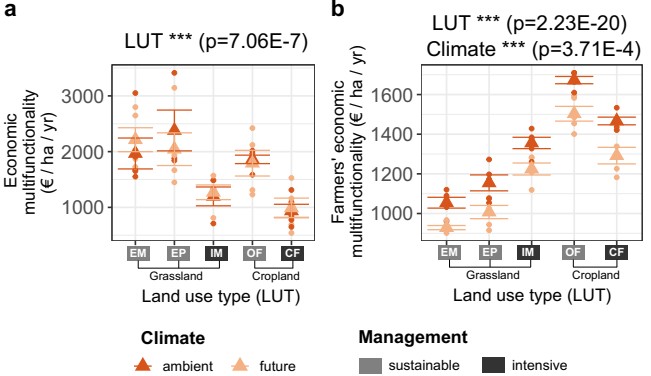

**Fig. 4 | Economic multifunctionality under different land-use types.** Economic ecosystem multifunctionality value **a** and farmers' economic value **b** as affected by five different land-use types (EM: extensive meadow, EP: extensive pasture, IM: intensive meadow, OF: organic farming, CF: conventional farming) for both ambient and future climate. Dots indicate the level of economic multifunctionality and farmers' economic value within the plots of the experiment (5 replicates for each LUT-Climate combination), triangles indicate the mean value of the respective LUT-Climate combination group. For statistical testing, F tests based on ANOVA (two-sided) without adjustments for multiple comparisons were used (numerator df: 4; denominator df: 32). Asterisks indicate a significant effect of the respective factor or interaction (*** $p < 0.001$; ** $p < 0.01$; * $p < 0.05$). Error bars represent the standard errors of the mean.

The only exception is ecological multifunctionality according to farmers' preferences, which puts highest weight on the provisioning service (food production), but very low weight (5%) on the regulating and supporting services that make up a big part of the overall economic multifunctionality. Contrary to weighting according to farmers, other weighting scenarios have a more balanced consideration of all ecosystem services assessed (Supporting Data File 4).

## The role of biodiversity for ecosystem multifunctionality

A linear regression model is used to analyse the relationship between soil biodiversity and ecological multifunctionality. Therefore, soil biodiversity-related functions are not included in the multifunctionality measure used for this analysis, but rather used as explanatory variable (calculated as soil multidiversity to integrate soil biodiversity data of different taxonomic groups[27,42]), and ecological multifunctionality is composed of only 11 ecosystem functions with equal weighting[6,27,42,43]. For the regression analysis, data for all different land-use types are included and analysed separately for ambient and future climate. Ecological multifunctionality has a significant and positive correlation with soil biodiversity for all climate types ($R^2 = 0.147$, $p < 0.01$). A marginally significant positive correlation between soil biodiversity and ecological multifunctionality is found under future climate ($R^2 = 0.14$, $p < 0.1$), but not under ambient climate ($R^2 = 0.039$, $p = 0.344$, Fig. 6). Notably, this difference is due to the fact that ecological multifunctionality is generally higher under ambient climate, but that ecological multifunctionality is particularly negatively affected by future climate in land-use types with low soil biodiversity. The correlation between soil biodiversity and ecological multifunctionality does not differ significantly between the two climate treatments (interaction term $\beta = 0.187$, SE = 0.157, t = 1.129, $p = 0.265$).

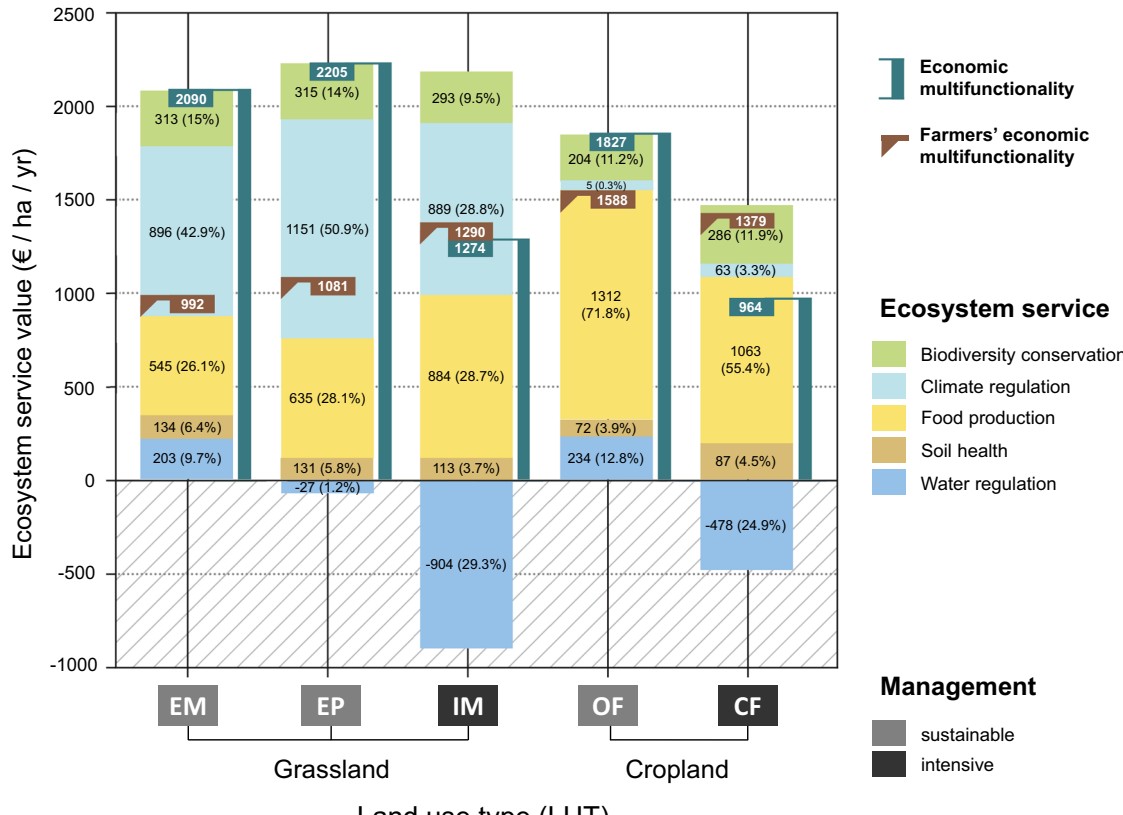

**Fig. 5 | Yearly ecosystem service provision in monetary units for different land-use types (EM: extensive meadow, EP: extensive pasture, IM: intensive meadow, OF: organic farming, CF: conventional farming).** Vertical bars indicate the economic ecosystem multifunctionality value (blue, including benefits for society at large) and farmers' economic value (brown, composed of food production and yield stabilising services biodiversity and soil health).

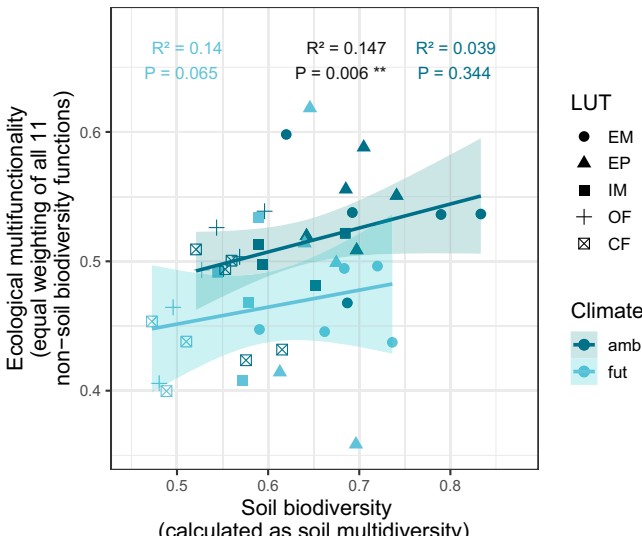

**Fig. 6 | Relationships between soil biodiversity and ecological multi-functionality (equal weighting of all 11 non-soil biodiversity functions) using a linear regression model across all land use types (LUT), separated for both climate scenarios.** The two regressions are represented by the two solid lines and accompanied by 95% confidence intervals (shaded areas). The three different R² and p values represent the regression model across all climate types (black text), the regression model for ambient climate (dark blue text) and the regression model for future climate (light blue text).

## Discussion

The present study contrasted ecological multifunctionality measures, based on averaging approaches with different weighting scenarios of ecosystem functions (according to different stakeholder preferences), to an economic multifunctionality measure based on the economic value of multiple ecosystem services for farmers and society at large. The selection of ecosystem services considered in studies of ecosystem multifunctionality will always be partly subjective (based on researchers' selection and data availability) and may exclude some ecosystem services (technically, weighting them with zero, here e.g. spiritual values / values of cultural identity). This is important to keep in mind, as the set of ecosystem services and functions considered will affect the assessment. Accordingly, we did our best to consider a representative set of key ecosystem services that includes all the benefits of major importance for stakeholders in the context of agroecosystems.

We find that future climate shows significantly adverse effects on ecological multifunctionality for most weighting scenarios and land-use types, supporting our expectation of a net negative effect of future climate on ecological multifunctionality[4,11]. The rather weak effect of the future climate for some land-use types (not affecting economic multifunctionality) can be partially explained by the relatively small temperature treatment in the GCEF with an increase of about +0.55 °C, which is in the lowest range of predictions at the time of the establishment of the experiment. Contrary to projections a decade ago, this future climate treatment may be a conservative estimate[44] of alterations towards the end of this century, with more recent predictions ranging between +1.1 °C to +5.5 °C, (reflecting all possible scenarios) and a mean of +3 °C[45]. The GCEF is among the few infrastructures that manipulate precipitation according to a realistic mean future scenario[44]. However, it can be assumed that the applied projected mean change of climatic conditions will have less severe effects on agroecosystems than the increased frequency of climate extreme years, which is an concomitant phenomenon of climate change[46]. As a consequence, we expect that future climate change may have even stronger detrimental effects on multifunctionality than presented in this study.

The present study further suggests that future climate effects differ with land-use types, which supports our hypothesis of significant differences in the resilience of certain land-use types[11,12] and biodiversity levels[19,47] to the effect of future climate. Ecological multifunctionality of different land-use types is differently affected by the simulated future climate, which aligns with other research showing a substantial spatial heterogeneity in the sensitivity of ecosystem service provision to future climate[11,12,48] and is in line with other studies showing grasslands being resilient to future climate extremes (severe droughts) irrespective of the management intensity[49]. In addition to the future climate scenario, land-use type and management regime could be shown to cause pervasive effects on multiple ecosystem functions and on multifunctionality in this study. The components of ecological multifunctionality (and of farmers' economic multifunctionality) that show the strongest response to future climate change are yield, flower abundance, and aboveground decomposition rate, all of which are strongly dependent on water availability that is decreased under the future climate treatment.

As data on flower abundance was only assessed at one point in one year, higher resolution phenological data over the whole growing period should be assessed for future research. Still, as the data was obtained at the beginning of June during the peak of aboveground standing biomass and highest flower abundance, it is assumed that our measure of flower abundance is representative of this ecosystem function. However, it is crucial to acknowledge the limitations of using only flower abundance as a measure for the cultural/recreational value of the ecosystem, as numerous aspects contributing to this value, such as the presence of sheep grazing on extensive pasture plots - commonly perceived as aesthetically appealing - could not be included in our assessment.

In both the ecological multifunctionality approach, which relies on stakeholders' preferences, and the economic multifunctionality approach, the monetary value of crop yields is used for the ecosystem service 'food production'. This means that, from a farmer's perspective, land-use types with the highest economic yield are preferred under both ecological and economic multifunctionality measures. Intensively-managed grassland and cropland show high multifunctionality for farmers, as they have high yields, while the dis-service nitrogen surplus, notably high in intensively managed land-use types, holds relatively less weight from a farmer's perspective in the ecological approach, and does not provide a benefit for the farmer in the economic multifunctionality measure. Those findings highlight the importance of private economic benefit for the farmers' preferences for land-use types.

Sustainably-managed cropland has the highest multifunctionality for farmers for both the ecological and the economic multifunctionality measures. The reason for the high monetary value of sustainably-managed cropland is that organically produced crops obtain high prices (subsidies not included), which outweigh the slightly lower mass yields (7%) compared to conventional crops, an effect also shown before[16]. We can contrast economic multifunctionality to alternative measures of economic profitability. According to economic theory, the private profitability of using land is reflected in the land rent (more precisely, the land rent captures the annuity on the maximal present value of private net benefits of land use[50]). Thus, land rent is a meaningful independent measure of economic value to compare with the economic ecosystem multifunctionality index. With a mean of 1,856 € / ha (grassland) and 1,396 € / ha (cropland), the yearly economic value of multifunctionality is substantially higher than the rent for land of similar quality in the region, where the experiment we obtained our data from is located (348 € / ha / yr for grassland, 574 € / ha / yr for cropland farmland in the Saalekreis district 2019)[51]. Partly, this is due to the fact that land rent is the net of the costs of farm management, while these costs (e.g., expenses for agrochemicals) are not included in the economic

multifunctionality index. Partly, this reflects that major contributions to economic multifunctionality are benefits to society at large rather than contributions to farmer's profits.

While future climate typically reduces ecological multifunctionality, the absolute reduction of ecological multifunctionality under future climate is highest for sustainably-managed cropland. Despite the high absolute reduction, under future climate, sustainably-managed cropland still maintains the highest ecological multifunctionality level of the five land-use types assessed for three out of four weighting scenarios. This demonstrates its overall benefit under both climate conditions. Under a farmer's perspective, ecological multifunctionality, which can be seen as the multidimensional benefit farmers receive from ecosystems, is strongly decreased under future climate. This is explained by the role of water as a limiting factor for plant growth during the growing season aligning with predictions of a net-negative effect of future climate on agricultural production for Europe[52,53]. By that, we expect farmers to be severely affected by climate change if adequate adaptation measures are not taken.

From a perspective of environmental conservation agencies, the intensively managed land-use types are less valued in the corresponding ecological multifunctionality measure. This is partly due to their high level of nitrogen surplus due to the application of mineral nitrogen fertiliser. This nitrogen surplus can leach into water bodies and groundwater when sufficient water is available outside of the growing season[54]. As plants cannot take up all available nitrogen if water availability during the growing season is too low, this might explain the increased nitrogen surplus under future climate observed in the present study. Due to the high soil water holding capacity, a deep root penetration and low precipitation within the area of the experiment[31], it remains unclear if nitrogen leaching occurs or whether the specific site conditions allow for complete denitrification.

Under a weighting according to preferences of environmental conservation agencies, ecological multifunctionality for sustainably-managed cropland is not different from multifunctionality for the sustainably-managed grassland types, and the sustainably-managed land-use types have in general higher levels of ecological multifunctionality, as they show no nitrogen surplus and high levels in soil health-related ecosystem functions. Although the level of nitrogen surplus is artificially set due to the application of mineral fertiliser in intensively-managed land-use types, this finding is in line with previous work showing that species-rich grasslands reduce nitrogen leaching from soils[55], support higher soil microbial biomass, activity, and diversity[56,57], and elevated decomposition rates[58].

Despite their lower economic value for farmers, grasslands showed higher overall economic multifunctionality than croplands due to their higher economic value for society at large. For both grassland and cropland, the sustainably-managed land-use types show economic multifunctionality values ~ 1.7 to 1.9 times higher than the intensively-managed types. Despite the expectation of a decrease in the ecosystem service provision under future climate[11,52], no significant decrease through future climate is shown here (3% decrease based on comparison of mean values of economic multifunctionality). This suggests that sustainable land use is a promising approach to maintain and promote the economic multifunctionality value of managed land also under a changing climate.

The contributions of the different ecosystem services to economic multifunctionality differs for the five land-use types. For grassland, the value for food production is slightly increased under intensive management due to the application of fertiliser. For cropland, sustainable management reaches higher monetary yields due to the higher sales prices of legumes and organically-produced crops, as shown before in ref. 16. The benefits in productivity in intensively-managed grassland are outweighed by the disadvantages of increased nitrogen surplus that significantly decreases the economic multifunctionality value of all intensively-managed land-use types. This is

highly relevant, given that surplus nitrogen and nitrogen contaminating groundwater is a critical issue in many intensively-managed regions of the world[59,60] and represents a threat to human health[61–63]. Still, economic multifunctionality is sensitive towards the social cost of nitrogen leaching, which depends on geographic characteristics and soil conditions of the respective site[64]. Under soil conditions that allow for complete denitrification before nitrogen leaches into water bodies, the negative effect of intensive management on multifunctionality might be smaller. While economic multifunctionality as calculated for the purpose of this study is based on a set of ecosystem services most relevant for the study area and ecosystem type (agricultural land), it does not represent the total economic value of the ecosystem, which would include all values attached to the ecosystem, including existence and bequest values, which are not directly related to current ecosystem functions.

Notably, the experimental plots were established on former agricultural fields (cropland) with disturbed soil conditions. For the two cropland types considered in this study (organic farming and conventional farming), the change in management due to the implementation of the experiment therefore was less pronounced. For the three grassland types considered in this study, the soil was gaining carbon due to the implementation of the experiment and the land-use change from cropland to grassland (-10% increase in total organic carbon over the first three years after the implementation of the treatments). This rate of soil organic matter accumulation is in line with other research showing a rapid accumulation of soil organic matter during the initial years following the conversion of cropland to grassland, which then diminishes in subsequent years[65,66]. As a consequence, the ecosystem service 'climate regulation' contributes a substantial proportion (39% to 62%) to economic multifunctionality of grassland, due to the carbon storing effect in the soil. This indicates that land managers would have quite different incentives if carbon sequestration would be rewarded economically (carbon farming). Given the historic use of the experimental site as cropland before the implementation of the treatments, the carbon storage effect is negligible for cropland types. This is likely because they might have already reached their soil carbon equilibrium, and their yearly carbon flows are about zero. The soil carbon content is highly interconnected with other ecosystem functions[67,68]. Due to the high dependency of the soil carbon content on the land management type, and its incremental increase over years[68] in response to the management practice, land-use change requires time to affect multifunctionality. By that, the detrimental effects of intensive land-use might increase in the future due to a loss of soil communities and soil related ecosystem functions[27,69,70]. While land-use change is very common in agriculture, the conversion from cropland to grassland for some treatments should be taken into consideration when interpreting our findings. Future research should aim to elucidate the equilibrium status of soil carbon in various pastures and meadows, explore alternative drivers of ecological multifunctionality beyond carbon storage, and investigate the interaction and the cause-effect relationship between soil organic matter and soil biodiversity with mixed-effect model experiments to further investigate the broader benefits associated with an increase of soil organic matter formation (e.g., water retention and nutrient cycling) resulting from different management practices.

The natural insurance values of soil health and soil biodiversity are based on their stabilising effect on aboveground biomass production, which aligns with previous research[38,71–73]. This insurance value for farmers depends on their risk aversion and might be smaller for less risk averse farmers (Fig. S5). Moreover, different findings of this study suggest that soil biodiversity is associated with ecosystem multifunctionality more generally, as indicated by a significant positive correlation between soil biodiversity and multifunctionality (equal ecosystem function weighting) ($R^2 = 0.147$, $p < 0.01$**, across all climate and land-use types). Although the experimental design does not allow

identifying causality, as this would require experimental manipulations of soil biodiversity, this observation goes in line with former research showing soil biodiversity to be a main driver of multifunctionality[27,29,43,69,70,74]. For grassland, for instance, the low multifunctionality level of the intensively-managed type goes hand in hand with a low (nematode-)biodiversity. This further aligns with previous findings showing that biodiversity loss is a main driver of multifunctionality reduction due to land-use intensification[6,29]. In our analysis, when separating by climate, the correlation between soil biodiversity and multifunctionality was only marginally significant under future climate ($R^2 = 0.14$, $p = 0.065$). While this contrasts with the findings describing an increase of (soil) multifunctionality with higher (plant) biodiversity in both ambient and future environments[19], it aligns with other findings describing a stronger effect of biodiversity on ecosystem functioning in more stressful environments[18] up to a certain threshold of environmental stress[75]. Even if only marginally significant, the correlation between soil biodiversity and ecological multifunctionality under future climate suggests that the role of (soil) biodiversity for multifunctionality might become even more important under future climate[76] and that promoting soil biodiversity might be important to mitigate the adverse effects of climate change. For interpretation of the results, it should be considered that the potential influence of other covarying factors on the relationship between soil biodiversity and multifunctionality were not explicitly accounted for in the experimental design, and the correlation is expected to be co-determined by the different management practices. Future research of the biodiversity-ecosystem multifunctionality relationship should include broader taxonomic variation in biodiversity assessments, including aboveground invertebrates.

By making use of combined ecological-economic multifunctionality models, our study sheds light on different aspects of ecosystem multifunctionality under different land-use types and climate change. While the ecological approach allowed the assessment of multifunctionality from the view of different stakeholder groups and the inclusion of cultural ecosystem services, the economic approach further allowed an assessment of multifunctionality on a societal level with a unified metric, grounded in economic theory. All ecological and economic approaches correlate strongly with each other (except farmers' ecological multifunctionality and economic multifunctionality). Under the ecological approach, sustainably-managed cropland shows the highest level of ecological multifunctionality under ambient climate, but is also the most sensitive towards future climate. Under future climate, however, sustainable-managed cropland does not show higher levels than the sustainably-managed grassland types. Utilising the economic approach, sustainably-managed grassland types show the highest levels of economic multifunctionality, also under future climate. As landscape multifunctionality strongly depends on land-use type heterogeneity, it is crucial to incorporate a mix of various land-use types to obtain high multifunctionality levels at large scale[15].

From the 1.2 million ha of agricultural land in Saxony-Anhalt (of which around 80% is managed as cropland, and around 15% is managed as grassland[77]), around 10.5% is managed organically (without heavy use of agrochemicals, corresponding to the definition of sustainable management used in this study). Although a detailed overview of the shares of the land-use types assessed in this study for Saxony-Anhalt is missing, this strongly indicates the need to shift to a higher share of sustainable land management, considering the higher level of both economic and ecological multifunctionality levels that organic farming shows for all stakeholder scenarios, also under future climate. Comparing farmers' economic multifunctionality under different grassland management scenarios reveals an economic multifunctionality provision being around 250 € / ha / yr higher for intensive compared to sustainable management, which can be seen as the yearly payment a farmer would need to receive, to switch from intensive to sustainable grassland management. This value is close to the real agricultural

subsidies of 267 € / ha / yr in Saxony-Anhalt, Germany, that farmers receive to switch from intensive to sustainable grassland management[78]. The fact that the share of sustainably-managed grassland is still low suggests that economic incentives need to be better designed to incentivise farmers to choose the management type that is preferred from a societal perspective.

Taken together, the present work highlights the risk of a significant decline in multifunctionality due to land-use intensification, climate change, and biodiversity loss, and the corresponding adverse value to humans. Therefore, we suggest to promote sustainable land management (especially the sustainably-managed grassland types extensive meadow and extensive pasture that are less sensitive to future climate) as well as to implement measures and incentives to increase soil biodiversity within agricultural areas fostering multifunctionality. Notably, this study also introduces an important conflict. Agricultural land is typically managed by farmers. However, those are the only stakeholder group considered in this study, whose multidimensional benefit obtained from multifunctionality was higher under intensive management (for grassland types) – at least in the short term –, which is underlined by the value gap between farmers' monetary benefit received from agroecosystems and their respective economic multifunctionality value for society. This further becomes evident when comparing the correlations between various multifunctionality metrics (Supporting Data File 4). Ecological multifunctionality based on farmers' preferences and economic multifunctionality at the societal level are the only multifunctionality measures that do not correlate with each other. This is because yield is dominant in the farmers' ecological multifunctionality index, whereas the economic multifunctionality index puts a high value on economic damages that farmers are not held responsible for. This inherent preference discrepancy between farmers (as land managers) and society (as beneficiary from the land) points to the need to regulate farmers' land management in order to achieve the optimal long-term outcome for society. As a consequence, we suggest providing incentives to make farmers choose the land management that is preferred from a macrosocial perspective and to adapt compensation schemes that currently put too little emphasis on sustainable management practices and environmental measures, working towards an economy incorporating external costs.

## Methods
### Experimental setup
Data was obtained from the Global Change Experimental Facility at the field research station of the Helmholtz Centre for Environmental Research in Bad Lauchstädt, Saxony-Anhalt, Germany (51°22'60 N, 11°50'60 E, 118 msl)[31]. This area is characterised by a sub-continental climate and predominantly west winds with mean annual precipitation of 489 mm (1896-2013), respectively 525 mm (1993-2013), and mean temperature of 8.9 °C (1896-2013), respectively 9.7 °C (1993-2013). The soil of the study site is a Haplic Chernozem representing one of the most fertile soils to be found in Germany[31]. The GCEF was designed for a simultaneous manipulation of land-use type (5 types: 2 cropland and 3 grassland types) and climate (ambient and future), using a fully randomised split plot experimental design that allows full-factorial combination of the climate and land-use types with 50 plots of approximately 400 m$^2$ each (5 replicates for each land-use type-climate treatment combination)[31]. The experiment was established in 2013 on a field formerly used as cropland and has been ongoing since then. Manipulation of climate and the establishment of land-use types started in 2014. Land-use types represent five agricultural management forms typically practiced in Germany: Conventional farming (CF) including a crop rotation with winter rape, winter wheat and winter barley and application of mineral fertilisers and pesticides; organic farming (OF) including legumes in the crop cycle every three years (alternating alfalfa and white clover) to replace mineral N fertiliser as well as only physical weed control without the application of

herbicides; intensive meadow (IM) that includes the sowing of a seed mixture (four different *Poaceae* species) and the application of mineral fertiliser at the beginning of the growing season as well as after the first, second and third cut; extensive meadow (EM) that includes the sowing of a seed mixture with 56 species (14 grass species, 10 legumes and 32 herbs) representing a wide range of plant functional types typically found in grasslands in Germany; and extensive pasture (EP) that includes the same seed mixture as EM and a grazing with sheep (~20 sheep grazing 24 hours per plot) which takes place three times a year (early spring, mid- to late spring, and mid of summer). The land-use types can be differentiated into grassland (IM, EM, EP) and cropland (OF, CF), and into sustainable (OF, EM, EP) and intensive (CF, IM) management. Sustainable management refrains completely from the application of agrochemicals (mineral nitrogen fertiliser and pesticides). Climate manipulation is reached with large, permanent steel constructions that cover each plot, equipped with mobile shelters, side panels and an irrigation system, whereas night temperature is passively increased via automatic closing of the shelters and panels from sunset until sunrise resulting in an 0.55 °C increase of daily mean temperature and a stronger increase by up to 1.14 °C in average in minimum temperature. Summer precipitation is reduced by ~20% via control of the roofs by a rain sensor. Precipitation in spring and autumn is increased by ~10% with an irrigation system. With this treatment, a consensus scenario across different models of climate change in Central Germany for the years between 2070 and 2100 is simulated. The control plots that are managed under ambient conditions are equipped with the same steel construction to exclude possible microclimate effects on the experimental setup. Before the start of the experiment, oat was sown on all plots in 2013 to homogenise soil conditions[31].

## Samplings and measurements

Measurements in this study were taken from distinct samples representing unique observations in the study area. During the years 2014 to 2020, plots were harvested with a combine harvester. Aboveground biomass of yield (dt / ha, for cropland differentiated into grain and straw yields) was measured after air drying which left the biomass with residual moisture of 14% (barley / wheat grains) and 9% (rape grains). Depending on the annual environmental conditions, for grassland, harvesting occurred up to four times per year. For the total productivity over the year, yields of all harvests are summed up for each plot. For extensive pasture, machine harvest was not practical, as plots were grazed with sheep. Instead, harvesting was done manually right above the soil (for each plot, four subsamples were taken and averaged). Here, total yield also considers the grazing uptakes of the sheep, measured as the difference between the biomass in four subsamples in sheep-excluding cages and the four subsamples in the sheep area. (Bio-)mass yield was converted to monetary yield based on producer prices (Table S10).

Soil samples were taken 2015 and 2016 in the beginning of September from the topsoil layer (0-15 cm). Twenty drillings were collected per plot, pooled together, and sieved at 2 mm. Total organic soil carbon contents (% of dry soil) were determined in duplicate via dry combustion using a Vario EL III C/H/N analyser (Elementar, Hanau, Germany)[79]. All measurements in the following were conducted in water baths at 20 °C in an air-conditioned laboratory at iDiv using an automated $O_2$ micro-compensation system[80]. Before the start of measurements, samples were kept at 20 °C for 5 days to adapt the soil microbial community to a constant and standardised temperature. Soil microbial biomass C (MBC) was measured by substrate-induced respiration, i.e., the respiratory response of microorganisms to glucose[81]. To saturate catabolic microbial enzymes, 8 mg glucose $g^{-1}$ soil dry weight was added as aqueous solution to the soil samples. The mean of the three lowest hourly measurements within the first 10 h (excluding the first 2 h) was taken as the maximum initial respiratory response (MIRR)—a period where microbial growth has not started. Microbial biomass (µg C $g^{-1}$ dry soil) was calculated as 38 × MIRR

($µl O_2 g^{-1}$ dry soil)[82]. While microbial biomass can be used as a proxy for the soil water availability (as microbes depend on an aquatic environment to survive and proliferate, and are positively affected by a higher soil moisture), it is also dependent on other factors such as temperature, soil pH, organic carbon availability or oxygen level.

Enzymatic activities (nmol $h^{-1}$ $g^{-1}$ dry soil) were determined in years 2015, 2016, 2017, 2019 and 2020 using 4-methylumbelliferone (MUF)-labelled substrates[83]. Three enzymes (cellulase, N-acetylglucosaminidase, acid-phosphatase) that are ubiquitous in most organisms and represent the carbon, nitrogen, and phosphorus cycles were measured as indicator for the rate at which microbes can decompose and process organic matter to provide nutrients that are accessible to plants (phosphorous, nitrogen). For the measurements, individual black 96-well microplates were set up for each soil sample. These plates included enzyme-specific substrates, MUF dilutions (at 1.25 and 2.5 µM) to calculate quench and extinction coefficients, as well as controls for substrate and soil suspension. Approximately 250 mg of fresh soil sample was then suspended in 50 ml of acetate buffer (50 mM, pH 5) for analysis. To disrupt soil aggregates, the soil suspensions were sonicated for 5 min, transferred to the prepared microplates, and incubated at 25 °C for 60 min. The addition of 30 µl of 1 M NaOH solution stopped the enzymatic reactions. Fluorescence measurements were conducted using an Infinite 200 PRO plate reader (Tecan Group, Männedorf, Switzerland) with excitation at 360 nm and emission at 465 nm. Enzyme activities were reported as the turnover rate of the substrate in nmol per gram of dry soil per hour.

Belowground decomposition rate that is closely linked to nutrient cycles and includes effects of soil meso- and macrofauna was measured 3-weekly from 2015 to 2016 using bait lamina PVC stripes (1 mm×6 mm x 120 mm, Terra Protecta GmbH, Berlin, Germany) with 16 holes of 1.5 mm diameter in 5 mm distance. The sticks were filled with a bait substrate that consisted of 70% cellulose powder, 27% wheat bran, and 3% activated carbon. The strips were inserted vertically in the soil that was prepared for insertion with a steel knife, just below the ground surface. The average bait consumption of 5 bait lamina stripes that remained in the soil for 3 weeks within one plot were, whereas holes were determined as empty (1), partly empty (0.5) or filled (0)[84]. Aboveground decomposition rate (microbes / microbes + fauna) was measured using litterbags (0.02 mm / 5 mm mesh size) filled with 12 g of air-dried oat plants (with stems and leaves), which were harvested as green plants on the study site in 2013 before the implementation of the experimental treatment. The litterbags were left on soil for two (summer and spring) or 4 months (winter) with a total of 7 measurement periods: period 1 (spring): 10.04.2015 – 4.6.2015; period 2 (summer): 4.6.2015 – 10.8.2015; period 3 (winter): 22.10.2015 – 8.3.2016; period 4 (spring): 8.3.2016 – 7.6.2016; period 5 (summer): 28.6.2016 – 31.8.2016; period 6 (autumn): 31.8.2016 – 30.10.2016; period 7 (winter): 30.10.2016 – 7.3.2017. In each period, a total of 200 litterbags (100 fine-meshed and 100 coarse-meshed) were placed randomly along a transect of 15 × 0.5 m into the 50 sub-plots. After the end of each period, the cleaned litter residues were dried at 70 °C for at least three days, and were weighted afterwards[85]. Soil mineral nitrogen ($NH_4^+$ and $NO_3^-$) content (mg / kg dry soil) was measured in 3-weekly resolution from 2015 to 2017 per flow injection analysis (FIAstar 5000, Foss GmbH, Rellingen, Germany). Therefore, 5 g of fresh soil was suspended in 20 ml of 1 M KCl solution, shaken for 1 h on a horizontal shaker and filtered through 0.45 µm cellulose nitrate filter (Sartorius Biolab Products, Göttingen, Germany)[79]. Mineral nitrogen deprivation (the removal of nutrients from the soil, in kg N / ha) was measured over the years 2016 to 2019 through an elemental analysis of the harvested plant biomass (for organic and conventional farming only for the machine harvest (cut 10 cm above soil), for intensive and extensive meadow also for the manual harvest (cut 3 cm above the soil surface), for extensive pasture only for the manual harvest). Plant material (dried at 70 °C for 48 h) was shredded and homogenised, a

subsample was milled to a fine powder, and appr. 10 mg of the finely milled plant material were weighted with an analytical microbalance (Cubis MSA 3.6 P, Sartorius AG, Göttingen, Germany) into tin capsules and measured with an elemental analyser (Vario EL cube, Elementar Analysensysteme GmbH, Langenselbold, Germany). Nitrogen stocks were calculated based on data on yield (dry biomass). Soil nematode biodiversity was measured in 2015 and 2016 (two measurements per year, in both spring and autumn). Seven soil subsamples per plot were taken using a steel corer (1 cm diameter; 15 cm depth), homogenised, sieved at 2 mm, and stored at 4 °C. Nematodes were extracted with a modified Baermann method[86], whereas for each plot approximately 25 g of soil was transferred to plastic vessels with a milk filter and a fine gauze (200 μm) at the bottom and placed in water-filled funnels. To ensure soil sample saturation and to ensure a connected water column throughout the sample and the funnel that allowed nematode migration from the soil through the milk filter and the gauze into the funnel, more water was added. After migration, nematodes gravitationally settled at the bottom of a closed tube connected to the funnel, and after 72 h at 20 °C, were transferred to a 4% formaldehyde solution, and were counted at 100 × magnification by using a Leica DMI 4000B light microscope. Identification was conducted at 400 × magnification. In order to identify the specimens, sediment material from the bottom of each sample was collected using a 2 ml plastic pipette. The collected sediment was then examined on temporary microscope slides. A minimum of 100 well-preserved specimens, chosen at random from the sample (if available), were identified up to the genus level for adults and most juveniles, or at the family level for juveniles[87]. Subsequently, the nematode taxa were categorised into trophic groups, including bacteria, fungal, and plant feeders, omnivores, and predators, and nematode Shannon diversity index was calculated using the R package 'vegan' for both spring and autumn sample, and averaged for each year[88].

Soil meso- and macrofauna diversity was measured in 2015 and 2016 (two measurements per year; spring and autumn). During each sampling event, three soil core samples (with a diameter of 6 cm and depth of 5 cm) were collected per plot to extract mesofauna, primarily consisting of Collembola and Acari. The Macfadyen high-gradient extractor[89] was employed for this purpose. Collembolans were identified up to the family level, while Acari were identified up to the order level, utilising a VHX-Digital microscope. For macrofauna, two soil cores (with a diameter of 16 cm and depth of 5 cm) were taken per plot. A Kempson extraction method[90] was employed, which involved gradually increasing the temperature over a span of 10 days. Macrofauna at the family level (Staphylinidae, Carabidae, and Formicidae), order level (Diptera, Araneae, Isopoda, Haplotaxida, Julida, and Psocoptera), or class level (Chilopoda, Araneae, Symphyla, and Gastropoda) were identified and Shannon index was calculated accordingly (separate for meso- and macrofauna, and for the spring and autumn samples), and averaged for each year[91].

## Calculation of nitrogen surplus

Nitrogen surplus was calculated on plot level as the soil mineral nitrogen content at time x plus the annual input of mineral nitrogen[31] minus the annual nitrogen deprivation through the harvest plus the soil mineral nitrogen content at time x + 1 year[92]. For x, the time between one year's harvest and the sowing / first fertilisation for the next year's harvest was chosen (end of July to beginning of August for cropland types; end of January to beginning of February for grassland types). The mean of two measures was used due to the variability of the nitrogen state due to weather conditions. Different datasets were unified (mineral nitrogen content in g / m² and in mg / kg soil) based on the assumption that the applied fertiliser spreads and accumulates in the upper 20 cm of the soil and that soil weight is 1350 kg/m³ [93]. Nitrogen outgassing and deposition were not considered here as they are opposing processes that can equalise each other[92]. While this calculation method is able to identify general trends regarding the

amount of nitrogen surplus after the end of the growing period, site-specific climate and soil conditions can strongly affect the magnitude and accuracy of the calculated nitrogen surplus. Therefore, it is crucial to interpret the results within the context of these specific conditions and recognise the potential for variability and inherent uncertainty inherent in the method.

## Calculation of soil multidiversity

To integrate the three different available datasets on soil biodiversity (data on three different organism groups: soil mesofauna, soil macrofauna, soil nematodes) into a proxy for the overall soil biodiversity level, the "soil multidiversity" was calculated as the average proportional Shannon-index biodiversity across the organism groups, (or respectively, size categories that usually contain certain taxonomic groups), normalised with the maximum observed level of Shannon-index biodiversity of the respective organism group to make sure that organism groups with different total species numbers are weighted equally[94].

## Calculation of soil health

Soil health was calculated as the average of the normalised level of the three ecosystem functions microbial biomass, enzymatic activity (calculated as the average of the normalised performance of the three ecosystem functions cellulase activity, N-acetylglucosaminidase activity and acid-phosphatase activity) and decomposition rate (calculated as the average of the normalised level of the three ecosystem functions belowground decomposition rate, aboveground decomposition rate (microbes) and aboveground decomposition rate (microbes + fauna)), as those ecosystem functions increase the long-term soil water retention and nutrient availability for plants[19,33].

## Assessment of flower abundance

A picture of each of the 50 plots was taken in the beginning of June 2023, when standing aboveground biomass and flower abundance was highest (camera model: Lumix DMC-FZ38, 12 megapixel with 4000 × 3000 pixel). Pictures were taken under sunny weather conditions, from the edge of the plots and from a height of 2 m, photographed at a 45-degree angle into the plots. An image analysis of the pictures was conducted to detect and quantify specific colour pixels indicative of flowers. The goal was to identify and analyse the prevalence of red, orange, blue, violet, white, and yellow colours, which are commonly associated with various types of flowers. Weeds with colourful flowers like *Papaver rhoeas* mainly occurred on the land managed sustainably (weed occurrence unwanted for organic farming, and part of the sown seed mixture for extensively managed meadow and extensively managed pasture), and was missing under intensive management, where the target plant mixture is too dense and vigorous to allow for the establishment of weeds. Pictures were taken three weeks after the first grazing of the extensively managed pasture plots, which, by that, show reduced flower abundance. While this represents the reduced aesthetic value of extensively managed pasture during the peak of the flowering season based on the flower abundance, it neglects the presence of sheep during grazing which can be seen as an aesthetic value, too. For the image analysis, in a first step, each image was edited manually using Gimp[95] to black out areas that could be misidentified as flowers, especially white sticks from other experiments on the plots. Each picture was then processed using the Python programming language[96] and the libraries PIL (Python Imaging Library), pandas, and os. The code was deposited in the Zenodo database under Creative Commons Attribution license[97]. The pictures were converted to the RGBA (Red-Green-Blue-Alpha) colour mode to facilitate pixel analysis and transparency adjustments. For each picture, a pixel-wise analysis was performed. The code iterated over every pixel, extracting the RGB (Red-Green-Blue) values. The extracted RGB values were used to identify pixels falling within predefined ranges corresponding to the

colours associated with flowers. Pixels were classified as red if their red component was above 200 and the green and blue components were below 100; as orange if their red component was above 200, their green component was above 100, and their blue component was above 50 but below 150; as blue if their red and green components were below 100 and their blue component was above 200; as violet if their red and green components were below 100 and their blue component was above 150, with the absolute differences between the red and blue components smaller than 50; as white if their red, green and blue components were above 200; and as yellow if their red and green components were above 200 and their blue component was below 100. The component thresholds were identified via a manual check of the component ranges of flowers within the analysed pictures. Detected flower pixels were marked within the images to visualise their locations. Pixels falling outside the identified colour ranges were marked with adjusted transparency to differentiate them from the flower pixels in the output picture. The total count of red, blue, white, and yellow pixels within each image was determined. These colour pixels were considered as potential flower candidates, as they corresponded to the typical colours found in flowers. The percentage of these colour pixels was calculated by dividing the colour pixel count by the total number of pixels in the image. The results, including the image filenames and the corresponding percentages of colour pixels associated with flowers, were recorded in a tabular format. A background noise of pixels falsely identified as white flowers could not be excluded (-1.37% of pixels as the mean of all CF and IM plots which in general did not contained any colourful flowers), but could be neglected due to the normalisation of the ecosystem function values based on the minimum and maximum observed value during the calculation of ecological multifunctionality.

## Calculation of ecological ecosystem multifunctionality

Ecological ecosystem multifunctionality ($egEMF$) was calculated with the averaging approach as weighted average of the levels of the different ecosystem functions $EF_i$ (according to their relative share of the respective ecosystem service) with $\alpha$ as a weighting factor, where $EF_i$ is calculated as a fraction of an actual value $X_i$ to two reference values (minimum ($X_{i,\min}$) and maximum ($X_{i,\max}$) observed value of the respective ecosystem function $i$)[3]. The first two years of the experiment were excluded, as many soil functions require time to respond to experimental treatments (see Table S1 for an overview of years used for different ecosystem functions).

$$egEMF = \sum_{i=1}^{N} \alpha_i EF_i \qquad (1)$$

**Calculation of ecosystem service preferences of different stakeholder groups based on survey data.** The ecosystem service preferences of the four different stakeholder groups most affected by the ecosystem type assessed in this study (agricultural landscape) and by the specific services this ecosystem type provides were obtained from survey data[41]. Those four stakeholder groups are farmers, local residents, environmental conservation agencies, and the tourism sector. A transformation process was necessary to harmonise the different sets of ecosystem services investigated in this study and in the study where the survey data was obtained from ref. 41. In a first step, ecosystem services relevant for the particular ecosystem assessed in this study (agricultural ecosystem) were identified in the survey data[41] (Supporting Data File 5). Preferences for a new ecosystem service 'food production' was calculated as the mean of the two ecosystem services 'food production (from crops)' and 'livestock production' to represent provisioning service preferences for both land use types considered in this study (grassland and cropland). As the land-use types considered in this study provide either food production (cropland types) or

livestock production (grassland types, ultimately delivering food production), the choice of averaging the preferences for 'food production' and 'livestock production' instead of adding them ensured that an overestimation of the relative importance of the ecosystem service 'food production' in the context of this study is prevented. In a next step, a proportional allocation method was applied to acknowledge the inclusion of one additional supporting service and one additional regulating service in this study. Therefore, we multiplied the percentage weighting allocated to the ecosystem service 'biodiversity' (classified as cultural service in the original survey data[41], classified as supporting service in this study) by 0.5, and allocated this weighting to both supporting services used in this study (biodiversity conservation and soil health). The value of 0.5 was used to balance the contributions of the existing ecosystem service of the respective service class and the newly included one considering their relative importance, avoiding disproportionately overemphasising either service, and maintaining a pragmatic representation of their combined significance. The same was done to include an additional regulating service (water quality) to the original regulating service (climate regulation / carbon storage) to ensure a fair distribution of stakeholder preferences across all supporting and regulating services. A normalisation process was conducted by scaling down the percentage weightings originally assigned to the services. This adjustment ensured that the newly included services, soil health and water quality, were fairly incorporated, while maintaining a total weighting of 100% across the expanded set of services. For instance, while in the original survey data for the subset of ecosystem services used in this study, a weighting of 44% (food production), 29% (biodiversity conservation), 16% (landscape aesthetics), and 11% (carbon storage) for farmers was applied, this resulted in a weighting of 38% (food production), 18% (for each biodiversity and soil health), 14% (landscape aesthetics), and 7% (for each climate and water quality). See Fig. S1 and Supporting Data File 5 for an overview of the preference transfer. While the original survey data[41] did only include ecosystem services that represent final benefits (provisioning, regulating, cultural services), we further included two supporting services in the presented study. While the original survey data[41] classified biodiversity conservation as a cultural service, we classified it as a supporting service. Although these methodological adaptations introduce a level of subjectivity in the weighting process that influences the final calculation of ecological multifunctionality, it aligns to overall trends regarding the preferences of key stakeholder groups for different services[2]. Due to the extensive data manipulation that was required to transfer the measured stakeholder preferences from the original survey data[41], the weightings employed in this study can be viewed as refined adaptation loosely based on the original data.

Weighting according to farmers' preferences gives less weight to ecosystem functions related to soil health and biodiversity and landscape aesthetics, but high weight to food production. Local residents and the tourism sector both show high preferences for landscape aesthetics. Local residents further show higher preference for food production. Environmental conservation agencies show high preferences for supporting and regulating services, and lower preferences for landscape aesthetics and food production.

## Calculation of economic ecosystem multifunctionality

Yield raw data was obtained as data on dry biomass and used to calculate the market value 'food production' based on current market prices (Table S10). The economic value of the ecosystem service 'climate regulation' (non-market value) was calculated by multiplication of the mean net carbon flux in the soil per hectare and year (total organic soil carbon mass at year x – total organic soil carbon mass at the beginning of the experiment in year 2013) * 3.66 (transformation factor from carbon mass to $CO_2$ mass based on the atomic mass of carbon and oxygen) with an accounting price of 195 € / t of $CO_2$. This accounting price is recommended by the German Federal Environmental Agency

for assessing environmental costs, and is meant to capture the social cost of carbon, i.e., the aggregate damages of emitting a ton of $CO_2$ into the atmosphere. Transformation of the percentage specification of total organic carbon in the soil mass into carbon mass per area was done based on the assumption of a carbon distribution in alifsols up to a depth of 30 cm[98] and a soil weight of 1350 kg/m³ [93].

The economic value of the ecosystem service 'water quality (non-market value)' was calculated by multiplication of the annual net nitrogen flux (kg nitrogen surplus/ha) with an accounting price capturing the social cost of excess nitrogen in agroecosystems. Due to the geological site conditions, especially the occurrence of slack water (surface water accumulated on an impermeable or less permeable soil layer)[99], a social cost value of 7.30 € / kg $N_r$ surplus is used as the social cost value for $N_r$ leaching into surface water bodies, again following the guidelines of the German Federal Environmental Agency for assessing environmental costs[100].

To investigate the robustness of the results, we calculated economic multifunctionality using alternative price scenarios, that included a social cost of $CO_2$ emission of 280 € / t[101], a market price of 90 € / t (based on the market value of $CO_2$ emission within the compliance market of the European Emission Trading System (ETS) in 2021), and social cost of nitrogen leaching of 1.9 € / kg (accounting price for nitrogen leaching into groundwater[100]). Prices for one ton of CO2 in the voluntary market were not considered in this study, due to their high fluctuation.

To avoid double counting, the expected yield-increasing effect of soil health was not included in its economic valuation. However, the yield-stabilising effect of soil health was included in its economic valuation, representing its insurance value[71–73]. The insurance value is also an important part of the economic value of biodiversity, on top of the effects on average productivity[102] due to its stabilising effect on direct use values (e.g., yield)[34,103,104].

The calculation of the insurance value of the ecosystem services 'biodiversity conservation' and 'soil health' (non-market values)[34] is based on evidence obtained from our experimental data that both soil biodiversity ($R^2 = 0.197$, $p < 0.01$**) and soil health ($R^2 = 0.272$, $p < 0.001$***) correlate with yield stability (based on linear regression models across all land-use and climate types), which is an underlying assumption so that an insurance value can be calculated.

The risk premium $RP$ of the ecosystem function aboveground biomass ('yield') is calculated on plot level as the mean value of the yield $\bar{Y}$ minus the certainty equivalent, with yield coefficient of variation $CV$ and risk aversion $r$, assuming that the yield is lognormally distributed. A value of 0.28 was chosen for the relative risk aversion $r$ as the risk aversion of a slightly risk averse farmer[105].

$$RP = \bar{Y} - \bar{Y}\left(1 + CV^2\right)^{\frac{-r}{2}}, Y \sim lnN\left(\mu, \sigma^2\right) \quad (2)$$

Risk premium was shown as a function of biodiversity $b$ and soil health $h$. A multivariate, polynomial regression model was used to calculate the response surface (Fig. S6):

$$RP(b, h) = x_1 + x_2 b + x_3 b^2 + x_4 h + x_5 h^2 + x_6 bh \quad (3)$$

Then, the insurance value $I_B$ of biodiversity at biodiversity level $b$ and soil health level $h$ is given as the difference between risk premium $RP$ at biodiversity level 0 and soil health level $h$ and the risk premium $RP$ at biodiversity level $b$ and soil health level $h$:

$$I_B(b, h) := RP(0, h) - RP(b, h) \quad (4)$$

Vice versa, the insurance value $I_H$ of soil health at biodiversity level $b$ and soil health level $h$ is given as the difference between risk

premium $RP$ at biodiversity level $b$ and soil health level 0 and the risk premium $RP$ at biodiversity level $b$ and soil health level $h$:

$$I_H(b, h) := RP(b, 0) - RP(b, h) \quad (5)$$

so that the total insurance value $I_{HB}$ of the two insurance providing ecosystem services is given as the difference between risk premium $RP$ at biodiversity level 0 and soil health level 0 and the risk premium $RP$ at biodiversity level $b$ and soil health level $h$:

$$I_{HB}(b, h) := RP(0, 0) - RP(b, h) \quad (6)$$

The economic ecosystem multifunctionality value *enEMF* was calculated by summing the values of the single ecosystem services, which are obtained by multiplication of the level of the respective ecosystem service $ES_i$ with the accounting price $p_i$ of the respective ecosystem service.

$$enEMF = \sum_{i=1}^{N} p_i ES_i \quad (7)$$

### Statistical analysis

Flower abundance was calculated using the Python programming language[96] and the libraries PIL (Python Imaging Library), pandas, and os. All other calculations and the statistical analysis were conducted using R 4.0.4[106] with the packages 'extrafont', 'ggplot2', 'ggpubr', 'lme4', 'lmerTest', 'car', 'multcomp', 'multcompView', 'stringr', 'dplyr', tibble, lsmeans, DescTools, and 'plot3D'. The dataset and code were deposited in the Zenodo database under Creative Commons Attribution licence[97,107,108]. Statistical analysis of the level of ecosystem functions and EFM was conducted by application of linear mixed-effect models to plot-level data, with the factors Climate, Land-use type, as well as their interaction as fixed effects. According to the split-plot design of the experiment, Mainplot nested in Climate was included as a random effect (R automatically detects the nesting of Mainplot in Climate) and used as the error term for the effect of Climate. For statistical testing, we used F tests based on Type III ANOVA and Satterthwaite´s approximation (two-sided). To assess the normality of the residuals obtained from the linear mixed-effects model, a Quantile-Quantile (Q-Q) plot was constructed. To further investigate the effects of land-use and climate treatment on multifunctionality, we performed a Tukey post hoc analysis using linear mixed-effects models in R. Where the ANOVA revealed only a significant effect of land use and climate, but no interaction effect, we constructed a modified model by removing the interaction term to test the effect of land use on multifunctionality with a higher power. If no interaction effect was shown, estimated marginal means were obtained for the variable 'land-use type' using the 'lsmeans' function. If a land-use type-climate type interaction effect was observed, estimated marginal means were obtained for the land-use type variable and the land-use type-climate type interaction using the 'lsmeans' function. Subsequently, pairwise comparisons were performed to compare the estimated marginal means and to identify significant differences using the Tukey adjustment method. To examine the relationship between a set of variables, we calculated a correlation matrix, performing a correlation test for each combination of variables using the 'cor.test' function in R. The correlation coefficient estimate and corresponding p-value were extracted from the test results.

### Additional information

Supplementary Material and Supplementary Data comprises information regarding individual ecosystem function responses to different climate and land-use type types, as well as additional Fig.s and tables.

## Reporting summary

Further information on research design is available in the Nature Portfolio Reporting Summary linked to this article.

## Data availability

The data that support the findings of this study have been deposited in the Zenodo database under Creative Commons Attribution license[108]. Source data are provided with this paper.

## Code availability

Code for data cleaning and analysis has been deposited in the Zenodo database under Creative Commons Attribution licence[97,107].

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

## Acknowledgements

We thank the staff of the Bad Lauchstädt Experimental Research Station and the GCEF (especially Konrad Kirsch) for the maintenance of the study area and Stefan Klotz and François Buscot for their role in setting up the GCEF. We thank Elke Schulz for the provision of data on microbial biomass, plant available nitrogen and total organic soil carbon. We thank Gabriele Rada for supporting the data visualisation. Constructive comments by three reviewers improved our manuscript. The GCEF is a large investment of the Helmholtz Association, funded by the Federal Ministry of Education and Research, the State Ministry of Science and Economy of Saxony-Anhalt and the State Ministry for Higher Education, Research and the Arts Saxony. Further financial support came from the Helmholtz-Centre for Environmental Research Leipzig-Halle and the German Centre for Integrative Biodiversity Research (iDiv) Halle-Jena-Leipzig, funded by the German Research Foundation (FZT 118, 202548816). N.E. also acknowledges funding by the German Research Foundation iDiv ([DFG]; Ei 862/29-1; Ei 862/31-1).

## Author contributions

F.S., N.E., and M.Q. conceptualised and designed the research; M.S. & H.A. designed the GCEF; M.S. is the scientific coordinator of the GCEF; M.S., T.R., R.Y., H.A., I.M., E.B., C.R., W.S.H., J.S. & M.C. collected the data; F.M. calculated flower abundance; F.S. synthesised the data; F.S. analysed the data; F.S. wrote the first draft of the manuscript; F.S., N.E. & M.Q. were the core writing team; all co-authors contributed to revisions of the manuscript.

## Funding

## Competing interests

The authors declare no competing interests.

## Additional information

[1]German Centre for Integrative Biodiversity Research (iDiv) Jena-Halle-Leipzig, Puschstr. 4, 04103 Leipzig, Germany. [2]Department of Community Ecology, Helmholtz-Centre for Environmental Research - UFZ, Theodor-Lieser-Str. 4, Halle 06120, Germany. [3]Department of Soil Ecology, Helmholtz-Centre for Environmental Research - UFZ, Theodor-Lieser-Str. 4, Halle 06120, Germany. [4]Institute for Biology, Leipzig University, Deutscher Platz 5e, 04103 Leipzig, Germany. [5]Department of Physiological Diversity, Helmholtz Centre for Environmental Research - UFZ, Permoserstr. 15, Leipzig 04318, Germany. [6]Institute of Biology, Martin Luther University of Halle-Wittenberg, Halle, Germany. [7]Institute of Biological Research, Branch of the National Institute of Research and Development for Biological Sciences, Str. Republicii 48, Cluj-Napoca, Romania. [8]Department of Economics, Leipzig University, Leipzig, Germany. [9]These authors jointly supervised this work: Nico Eisenhauer, Martin Quaas. ✉e-mail: nico.eisenhauer@idiv.de

