## [Peer Review File · Nature Communications]

Reviewers' Comments:

Reviewer #1:

Remarks to the Author:

This paper describes the results of an experimental study in which the response of multiple ecosystem functions (which are also used as proxies of services) to various agricultural land management and climate regimes is measured. These multiple response variables are summarized in multifunctionality measures which summarize the overall performance of the system from both ecological and economic perspectives.

The overall experiment set up is impressive and the paper is good; it contains several aspects which I like a lot, in particular the treatment and linkage of functions and services shown in the first figure and the combination/comparison of both ecological and economic measures of ecosystem performance; whole ecosystem performance/multifunctionality is definitely given more thought than is the norm in this literature. I also found some of the valuation approaches interesting. Overall, it is a strong study that I enjoyed reading.

Unfortunately, at present a strong take home-message doesn't really manifest, however. This is due to weak framing of the original question and lack of really in-depth analysis and insight in any particular area. It's good, but it feels like a 'standard paper' - quite descriptive, rather than having a 'narrative', and lacking a very clear central point. The rationale needs to be stronger as reads a bit like 'it was done because it could be with the set up, and hadn't been done before'. A stronger rationale and framing of what follows in its light would improve the whole paper and help focus additional, more in-depth analyses and insight/discussion.

I don't think this is a fatal flaw but a reasonable amount of additional work is required before the paper is of the standard I'd expect from Nat Comms. ...also line numbers would have been useful!

Specific comments:

P2- preferences measured or estimated? TEV- includes all services or some omitted? (in which case its not the total)

P3

Allan 2015 reference- some more recent references may be better here as they use real stakeholder data (e.g. Linders et al 2021, Neyret et al 2021, 2023)

For biodiversity as a both a driver and response see Mace et al 2012 TREE

P4

I don't think the knowledge gap is particularly clearly communicated in this section- it should be something more than 'it hasn't been done before' - what additional insights will it bring, what are current approaches failing to do that the approach used here does? I would argue that the ecological approaches are complementary as they potentially represent different aspects of the system- e.g. current and long term value depending on how they are measured and cover a more complete valuation of the system when combined. Say what each can miss etc.

P5

Is the conservation of soil nematodes and meso- / macrofauna the part of biodiversity that people value in its own right? I would say its value is probably more acknowledged as a supporting service, with valuation of biodiversity 'in its own right' mostly belonging to pretty plants, and particularly vertebrates that are popular with the public. These organisms are great but of niche interest in the popular imagination.

P6

I really like Fig. 1- its nice to see this type of linkage and relationship between functions and services being considered properly. However, the array of functions and services is not complete (compare the list here to the 18 NCP of the IPBES framework for example) and the inclusion of other services/values- e.g. cultural values may alter the outcome. This should be acknowledged

(from a technical standpoint the unmeasured things are effectively weighted/valued as zero in both measures) and their exclusion may alter the outcome- what big NCP are missing here?

For the weighting, this is important as it massively alters the outcomes (e.g. see Allan 2015 or Linders 2021 People and Nature) and should be presented /made transparent – also is this real stakeholder data or assumed values/priorities?

P7

I see Biodiversity is represented as an insurance value- that seems fine, but contradicts the multiple 'values argument' presented earlier (see p5 comments) – please clarify which aspect of biodiversity is being valued, and which are not.

The analyses described at the end of the intro are not clearly articulated-. Why threshold approach, how were individual functions tested for?

Table 2 I guess equal ecosystem service weighting weighs the functions according to their relation to the services – e.g. the biodiversity variables are 1/3 of 20% - is that right? Please clarify in main text as its needed to understand the results

Figure 2 and results: a bit of talking through would be good here as help show which of these results are more meaningful and what to take from each. Similarly, clearer communication of the scenarios/weightings and what they represent earlier in the paper, and clearer articulation of the research question would help highlight the most important messages.

P10 – would be interesting to see how sensitive the economic valuation is to changes in prices etc (carbon value seems very high!). How much of this value is market and non-market? What's the market value multifunctionality?

P10- I've not dug into the methods yet but is it right to compare total value to rental rates? The agricultural yield values are presumably that of the annual crop/fodder, but the carbon cost is a one off- the field doesn't store that amount every year, so the comparison is somewhat misleading. (reading on I now see It Is a flux rate- make clear that all the values are some sort of annual yield here)

Figure 3 is a cool graph, but the farmer seems to be getting money for regulating services in their value- is that really happening?

Figure 5. Steepest, and positive slopes around t50 is something of a statistical null expectation when using threshold multifunctionality measures- see van der Plas 2016 Nat Comms and Gamfeldt and Roger 2017 NEE, though these two papers differ somewhat in their explanation of such effects This should be acknowledged, or better, shown how much it deviates from the null expectation.

P14- it would be good to see which functions/services are driving the decline in EMF under climate change, this should be highlighted and discussed. Some key points from the appendix could move up to illustrate this.

P15- Not always clear in the discussion if it is the economic or ecological multifunctionality that is being discussed (or both)

P16- I now see that this is a disturbed soil that is gaining C- this won't be the case for many pastures and meadows that are much closer to equilibrium- this is acknowledged but in a wooly way and this whole section is rather jumbled, vague and hard to understand. Please rewrite to make more specific points.

P17- the closing paragraph makes sense but says nothing particularly new- i.e. much work has already shown that non-market values need to be considered and included if nature is to be protected. Further, this is already reflected in part in for example, agri-environment schemes that effectively pay farmers to protect non-market values of farmland. What would be more interesting

is to see how much more the farmer would need to be paid, and for what, in order for them to make the socially beneficial decision. Also, what would the yield cost be to de-intensify and where would the extra food come from? How sensitive are these options to climate change? The discussion, and paper in general, would greatly benefit from going deeper on these points as the conclusions are rather generic at present, and this may make for a stronger framing of the paper.

I was also missing a discussion of what the two approaches showed us that was different, or the same, and what additional insights using both approaches give us. Mixed multifunctionality approaches- that combine ecological and economic measures (or multiple values in the same measure) are also starting to appear- e.g. Neyret et al 2023 Nat Sust. – how does the current approach relate to these?

This paper, as well as Neyret et al 2021 also combine different land uses to see how multifunctionality manifests at larger scales. Such papers highlight the limits of what we can say about identifying the best way of achieving local multifunctionality at small scales- is there some mix of the proportions of the land uses given here that would be the 'best' for all? I acknowledge it's a different question to move to this scale but this point may be worth exploring or discussing in some way.

Methods

P18 'representing all forms of vegetable life'- strange line and probably not true unless it contains the full range of grasses, herbs, lichens, mosses etc.

p19- justification of microbial biomass as a water measure could be clearer

p19- which three enzymes

p20- deprivation? (you mean concentration?)

p19-20- the methods are a bit too brief here and need to be expanded- sometimes even the basic name of the method is not given, or in the case of the soil biodiversity the taxa measured and how- to what level of taxonomic resolution where the soil organisms measured and how?

p20- the nitrogen surplus method is somewhat coarse with plenty of room for error in almost every part of the equation. The uncertainty should be acknowledged.

P20 – multidiversity- please say which groups went into this measure- plants too?

P21- its still a mystery as to what the weightings were, and where they came from- the detail can be in the SI but the basics should be here

P22- it would be nice to see some sensitivity check of the results to these values- the prices are always going to be somewhat arbitrary, and its also interesting to see how a valuation change can alter the outcome of what the 'best' option is, especially as these prices fluctuate over time in reality.

8.1 There is some important info here that would help make sense of the more synthetic measures- I would suggest saying which were the main responses/variables driving summary measures a bit more in the main paper to help guide the reader through what is going on.

Figure 7. Real data on stakeholder priorities in Germany show a far more even spread of priorities than is seen here- they tend to acknowledge each other's needs a lot more than you might expect (see Peter et al 2022 People and Nature). Some discussion of this or adjustment of the weightings in a sensitivity analysis if going deeper would be good. The fact that these are idealized, researcher defined priorities, should also be acknowledged in the main paper.

Reviewer #2:

Remarks to the Author:

The manuscript explores how land-use type and climate change affect ecological and economic multifunctionality using a 7-year manipulated field experiment simulating five common land-use types (three grassland and two farmland types either sustainably or intensively managed) and future climate in a site. The authors concluded that sustainable land management enhances ecological and economic multifunctionality under both ambient and future climate. Overall, this

work was of important value considering that its experimental design can help us simultaneously examine the effects of two major drivers--land use and future climate change on multiple ecosystem service and functions, which is less considered in previous studies. (I cannot give specific detailed comments due to lack of line number in the whole ms)

My major comments and concerns about this ms:

1. I don't think novel of this study lies in comparing sustainable land management with intensive management effects on EMF, because it is widely accepted that sustainable land management can offer more ecological service functions than intensive land use which only consider short-term high production functions. Instead, the novel of this work should focus on the interaction of land use and future climate, I therefore more wonder which land use will be confronted with great risk in offering ecological service functions in future climate changes. Unfortunately, the author showed that the interaction effects of LUT and climate was non-significant. I noted these results are from overall analyses including all five land use types (table 2, 3). However, this five land use types include two categories: grassland and farmland, and each included sustainable and intensive management. In your statistical models, whether these factors were considered? Grassland is greatly different from farmland, whether you attempted to conduct these interaction analyses in grassland and farmland, respectively, when you compare sustainable and intensive land management in future climate change. In your current results I cannot find these analyses. Moreover, whether you analyze the interactive effects of climate change and land use based on provisioning service, regulating service, and supporting service, respectively? Maybe these analyses could obtain new important findings.

2. For this work, another important and novel point was the considered economic measure of EMF value, especially for those economic value of ecological service provision excluding food production, which is less evaluated in previous EMF research. Future land use not only should consider its production value also include its economic cost or value from the other ecological service provision. So, this work is of important value to help guiding government to evaluate land use, also farmer to trade off how to use land more economically. I therefore suggest author more highlight this point in main finding and conclusion.

3. This study also analyzed the biodiversity-EMF relationship. However, I don't think these analyses are necessary in this work. Moreover, in this biodiversity-EMF relationship analysis, plant diversity seems not be included in this multidiversity index (I was not completely sure for this, because many data analyses method are not clearly introduced in this ms), while role of plant diversity in maintaining EMF cannot be neglected. I also wonder whether plant diversity was included in your ecological and economical multifunctionality? Because plant diversity conservation in land use should also be very important. According to Fig1, this seems not be included. Again, why do you use different EMF approach in analyzing biodiversity-EMF and land use/climate effects on EMF, one for threshold approach, the other for averaging approach. Both approaches should be examined in all analyses to solid these results. Moreover, I also noted that all functions except yield included in EMF index are about belowground soil functioning, so the current EMF index was more represent for soil EMF.

The other comments:

1. I am missing some important details on the methods. For example, although the authors showed that there were 50 plots, and each approximately 400m². How many replications for each treatment? And for the extensive pasture (EP) that includes grazing with sheep (~20 sheep grazing 24 hours per plot) which takes place three times a year (early spring, mid- to late spring, and mid of summer), are the large steel constructions for climate manipulation is removed in the grazing days? Also, for the samplings and measurements, all the samplings were conducted from 2014-2020, and you have the 7-year data? All the plots were harvested, or there have some sampling quadrats? How many sampling you have in total? There was no sampling replications? In addition, I noticed that you have mention the subplots and subsamples, what are these mean? soil sampling depth? These need clarification. In general, the quality of the current paper can only be fully assessed once these details are fully understood.

2. I also concern about the data analysis in this manuscript. The statistical analysis section is very brief currently, as the authors only provided the R packages for the analysis. I would also like to see more details on the actual statistical analyses. Do you use the multi-year data, how do you treat these data? Only use the mean, or conducted the repeated-measure ANOVA or the linear mixed effects models? Also, it seems that the analysis about the interactive effects of land-use type and climate on EMF is using the Type III ANOVA Satterthwaite's method in accordance with

the split-plot design of the experiment? Do this analysis using the plot level data? Do they include random terms? Based on the figure 2, 4, dots indicate the multifunctionality level within the plots of the experiment, triangles indicate group means, it seems there are many plots in each treatment. What about the sample size? Why table 2, 3 showed that the interactive effects of land-use type and climate on EMF was analysed by general linear regression model? So, how do the used models differ from the ANOVA analysis? Moreover, the authors have not provided the method for the biodiversity–multifunctionality relationship analysis. Regression analysis? Person correlations? Linear mixed–effects models? For this analysis, data in all the land-use types are included? And it was analysed in the ambient and future climate, respectively? Again, this does not mean that the analyses are wrong, but more details on the analyses should be provided.

3. I cannot understand why author analyzed EMF for all functions and also for all services In Fig 2? what's the calculating difference between EMF equal weighting of all functions (a) and all services (b)? According to fig 1, ecosystem services are also from these functions, it seems that they are from the same set of functions data.

4. The manuscript still needs a few improvements in terms of content and structure, especially for the introduction section. As the introduction nicely starts with ecosystem multifunctionality (EMF), however, the EMF has been widely investigated, it would be nice if they could point out the importance of considering both the ecological and economic multifunctionality. Not only show that here we do this. Then the authors introduced the land use, climate change, as well as the biodiversity as the drivers of EMF, however, all these factors included in the second paragraph, reads confusing and loss strength. It also leaves a reader wondering about what is the real novelty and significance which is relevant for publication in this journal. All the sections are not well-connected currently. These contents in the introduction need to be logically re-organized. Another key point is about the experimental duration, the authors should emphasize the multi-year data, which is important metrics of this study, as many studies using the one-year data, there may be the time-lag responses of ecosystem to land-use and climate change. In addition, I think that the discussion section could be shortened and improved. The discussions in each paragraph also are not well-connected currently.

Lastly, there was not line number in whole ms, it's very trouble when comment some specific points, so I strongly suggest adding line number in new revision ms.

Reviewer #3:

Remarks to the Author:

In the manuscript « Sustainable land management enhances ecological and economic multifunctionality under ambient and future climate” the authors investigate the impact of a crossed land use and climate change experiment on a range of ecosystem multifunctionality metrics, focusing on five main services: food production, climate regulation, water regulation, soil health and biodiversity conservation. They also propose an economic approach to multifunctionality to assess the economic impacts of said changes on society at large. They show that in general, MF would decrease in future climates, especially when considering farmers’ priorities, and less so when all services are considered equally (equal weight or environmentalists’ preferences).

This study addresses an important question for agro-ecosystems management and functioning in face of climate change, and the analyses seem sound. However, it would benefit from a tighter framing of the research questions and their corresponding metrics and analyses to address. As it is, some analyses are not fully justified or introduced. There are also a number of limitations regarding the calculation of multifunctionality that should be acknowledged.

Main comments

1. The authors provide a range of analyses, assessing how multiple multifunctionality metrics (7 in total) respond to climate and land use. However, why exactly each metric or analysis was selected is unclear. For instance, it might be unnecessary to show both the equal weighting of function AND the equal weighting of services, which adds to the overall complexity, unless more justification is provided on why each of these metrics is important. The other analysis could be moved to the supplementary material.

Besides, the threshold analysis comes a bit as a surprise at the end, and is currently a bit disconnected from the rest of the paper (e.g., why test climate and not land use if the rest of the paper tackles both issues?). The topic on climate impact on BEF relationship is introduced in the introduction, but explicitly stating what you're going to test (e.g. does climate change affects the strength of the relationship?) might help.

2. There are also some limitations that should be acknowledged, or clarified, regarding the calculation of multifunctionality.

- Services used: how did the authors choose the services? Do they really represent the most prioritised services required by local stakeholders? Would the inclusion of other services (eg cultural services) affect the results?
- Besides, it should be made clearer that the priority scores are made-up scenarios, not based on any social-ecological data (here the justification is limited to a citation of Manning et al 2018, but the scores used are different anyways). This, together with the fact that essentially only two stakeholder scenarios are considered (farmers and environmentalists) calls for
- Biodiversity: from what I understood, the biodiversity measures include essentially soil fauna. This is alright for using biodiversity as a function, or as a driver of multifunctionality. But I'm not sure it works as a service, because when people value biodiversity protection they rarely think about soil fauna, but rather plants, birds, etc. I suspect not much can be done about that in the manuscript, except acknowledging it.
- Soil health/food production: usually, ecosystem services multifunctionality (as in Manning et al. 2018) only includes final benefits, ie the benefits that directly benefit people, but not regulating services. This is to avoid double counting (e.g. pollination and the boost in food production provided by pollinators). There might be a similar problem with soil health here, which underpins food production and water regulation. Is there a risk that this would affect the results?

As a smaller comment, the terminology would benefit from being simplified, along with homogenised acronyms (if they are really needed). For instance, "economic multifunctionality" could be called "overall/total economic value" to homogenise it with the "farmers' economic value", which would remove the need to write "ecological ecosystem multifunctionality" for the other metric. Some acronyms are also only used towards the end of the manuscript (FEV, eEMF) – are they really needed? If so it would be good to introduce them earlier on and use them consistently.

I also have several minor comments on the manuscript. As the pdf I received did not have line numbers, I'm attaching a commented pdf instead.

Report how we have revised the manuscript

Sustainable land management enhances ecological and economic multifunctionality under ambient and future climate

For Nature Communications in response to the reviewers' comments

We are grateful for the constructive comments by the editor and the three reviewers. We feel that revising the manuscript and addressing these comments has made it much more convincing. In the following, we explain all changes we have made in response to the review reports. To this end, we reproduce the reviewer comments and provide the respective answers in *blue*. We have added numbering to the report by the second reviewer to facilitate referencing.

Our main revisions have been the following:

1. As proposed by reviewers 1 & 3, we included an additional ecosystem function (percentage flower cover) approximating an additional ecosystem service (landscape aesthetics) to represent a further ecosystem service class (cultural service).
2. As proposed by reviewer 1, we adjusted the ecosystem service weighting scenarios based on stated preferences of four different stakeholders (farmers, locals, environmental conservation agencies, tourism) derived from survey data (Peter et al., 2021), to replace the ad hoc estimated preference weightings. Further, we included two additional ecosystem service weighting scenarios based on stated preferences of two additional stakeholders (locals, tourism) to replace the equal ecosystem function and equal ecosystem service weighting scenario.
3. As proposed by reviewer 1, we calculated the optimal land use type composition as the land use type composition that maximises ecological ecosystem multifunctionality egEMF for all stakeholders.
4. As recommended by all reviewers, we clarified the research question and strengthened the main message around the climate effect and the land use type - climate interaction effect and around the benefits of mixed economic-ecological multifunctionality approaches.
5. As recommended by reviewers 2 and 3, we dropped the analysis of the biodiversity-EMF relationship with the threshold approach.
6. We identified optimal land use type composition scenarios (maximum community-level ecological ecosystem multifunctionality levels, or maximum ecological ecosystem multifunctionality equity across different stakeholder groups, respectively).
7. We included clearer definition and differentiation between ecological ecosystem multifunctionality (egEMF) and economic ecosystem multifunctionality (enEMF), see definition box.

Responses to Reviewer #1:

This paper describes the results of an experimental study in which the response of multiple ecosystem functions (which are also used as proxies of services) to various agricultural land management and climate regimes is measured. These multiple response variables are summarized in multifunctionality measures which summarize the overall performance of the system from both ecological and economic perspectives.

The overall experiment set up is impressive and the paper is good; it contains several aspects which I like a lot, in particular the treatment and linkage of functions and services shown in the first figure and the combination/comparison of both ecological and economic measures of ecosystem performance; whole ecosystem performance/multifunctionality is definitely given more thought than is the norm in this literature. I also found some of the valuation approaches interesting. Overall, it is a strong study that I enjoyed reading.

Response: We appreciate this very supportive feedback!

Unfortunately, at present a strong take home-message doesn't really manifest, however. This is due to weak framing of the original question and lack of really in-depth analysis and insight in any particular area. It's good, but it feels like a 'standard paper'- quite descriptive, rather than having a 'narrative', and lacking a very clear central point. The rationale needs to be stronger as reads a bit like 'it was done because it could be with the set up, and hadn't been done before'. A stronger rationale and framing of what follows in its light would improve the whole paper and help focus additional, more in-depth analyses and insight/discussion.

I don't think this is a fatal flaw but a reasonable amount of additional work is required before the paper is of the standard I'd expect from Nat Comms. ...also line numbers would have been useful!

Response:

To address this concern by the reviewer, we implemented several changes:

- In the revision of the manuscript, we clarified our research question and narrative around the future climate effect and the ability of different land use types to mediate the effects of future climate, as well as on the observed land use-climate interaction effect.
- Further, we put more emphasis on the complex and unique ecological design of the experiment, that allows to investigate the effects of climate change on EMF, which was not assessed in this form before. Moreover, this design allows to simultaneously examine the effects of two major drivers--land use and future climate change on multiple ecosystem services and functions, which have rarely been considered together in previous studies.
- We further clarified the need for mixed ecological-economic approaches when calculating EMF, as they overcome the shortcomings of each other (e.g., the shortcomings of economic approaches when assessing cultural ecosystem services, or their inability to include equity of EMF across stakeholders) (e.g., lines 68 to 75).

Specific comments:

P2- preferences measured or estimated? TEV- includes all services or some omitted? (in which case its not the total)

Response:

- Within the revision of the manuscript, we now use measured stakeholder preferences from survey data (Peter et al., 2021), which we clarified throughout the text (see e.g., abstract line 33, introduction line 179/180, methods lines 1082 to 1111).
- As not all ecosystem services were considered for this study, we refrain from using the term “TEV - total economic value”. Instead, we use the term “enEMF - economic ecosystem multifunctionality” and included a Definition box (line 77). We discuss the fact that not all ecosystem services could be assessed / monetised due to lack of data availability (e.g., line 150b).

P3 - Allan 2015 reference- some more recent references may be better here as they use real stakeholder data (e.g. Linders et al 2021, Neyret et al 2021, 2023)

Response:

- We included important findings of Neyret et al. 2023 regarding the benefits of mixed ecological-economic multifunctionality approaches and on identification of optimal land use type composition scenarios.
- We adjusted ecosystem service weightings according to a recent study investigating stakeholder preferences for landscapes based on a survey in Germany (Peter et al., 2021), which was used in Neyret et al. (2021 , 2023) (see Methods section for the adaptation of stakeholder preferences to acknowledge differences in the set of services assessed in this study and in the study of Peter et al. (2021)) (see e.g., abstract line 33, introduction line 179/180, methods lines 1082 to 1111).

For biodiversity as a both a driver and response see Mace et al 2012 TREE

Response:

- We included the reference.

P4 - I don't think the knowledge gap is particularly clearly communicated in this section- it should be something more that 'it hasn't been done before' – what additional insights will it bring, what are current approaches failing to do that the approach used here does? I would argue that the ecological approaches are complementary as they potentially represent different aspects of the system- e.g. current and long term value depending on how they are measured and cover a more complete valuation of the system when combined. Say what each can miss etc.

Response:

- We clarified the knowledge gap and research question
- We clarified the necessity of using both ecological and economic multifunctionality measures (see lines 68 to 75); We clarified the insufficient data / studies on how climate

change affects ecosystem multifunctionality, and clarified the valuable insights the complex ecological design used within the experiment we obtained our data from can bring (lines 120, 68-75)!

P5 - Is the conservation of soil nematodes and meso- / macrofauna the part of biodiversity that people value in its own right? I would say its value is probably more acknowledged as a supporting service, with valuation of biodiversity 'in its own right' mostly belonging to pretty plants, and particularly vertebrates that are popular with the public. These organisms are great but of niche interest in the popular imagination.

Response:

- Although it has been shown previously that the highest part of the economic value of biodiversity lays within biodiversity's stabilising role for other ecosystem functions and services (Farnsworth et al., 2015), some people value biodiversity more in its own right (biodiversity as a good, Mace et al. 2012).
- Accordingly, we valued the stabilising effect of biodiversity as part of the economic EMF index, whereas we included soil biodiversity in the ecological EMF index only. Although people appreciate soil biodiversity for its beauty and out of curiosity (Philips et al. 2020), we were not able to include a corresponding economic value in the economics EMF index due to lack of corresponding data for our study region.
- We acknowledge the fact that soil biodiversity is not the most intuitive part of biodiversity that is valued by people, but still has intrinsic value to people (Philips et al., 2020) (e.g., line 147).

P6 - I really like Fig. 1- its nice to see this type of linkage and relationship between functions and services being considered properly. However, the array of functions and services is not complete (compare the list here to the 18 NCP of the IPBES framework for example) and the inclusion of other services/values- e.g. cultural values may alter the outcome. This should be acknowledged (from a technical standpoint the unmeasured things are effectively weighted / valued as zero in both measures) and their exclusion may alter the outcome- what big NCP are missing here

For the weighting, this is important as it massively alters the outcomes (e.g. see Allan 2015 or Linders 2021 People and Nature) and should be presented /made transparent – also is this real stakeholder data or assumed values/priorities??

Response:

- While it is hardly possible to include all ecosystem services in an ecosystem assessment (especially considering the ~100 ES categories proposed by the CICES Common International Classification of Ecosystem Services), we address this comment by now acknowledging cultural services by the inclusion of the ecosystem function “% flower cover” representing the ecosystem service “landscape aesthetics”.
- We further added a paragraph (line 550-555) acknowledging the difficulty to include all services that an ecosystem provides (due to data availability), mention which service blocks are missing (especially spiritual / cultural heritage, sense of place and identity),

cognitive development and learning, and clarified that this technically represents a weighting of zero in both the ecological and economic multifunctionality measure.

- Weighting: see reply to earlier comment, based on measured stakeholder preferences (survey data)

P7 - I see Biodiversity is represented as an insurance value- that seems fine, but contradicts the multiple 'values argument' presented earlier (see p5 comments) – please clarify which aspect of biodiversity is being valued, and which are not.

Response:

- See comment before.

The analyses described at the end of the intro are not clearly articulated-. Why threshold approach, how were individual functions tested for?

Response:

- As proposed by reviewer 2 & 3, we dropped the threshold approach, as it is not necessary to investigate the biodiversity - EMF relationship. To investigate relationships between ecosystem functions / ecosystem services and EMF, we use linear regression models.

Table 2 I guess equal ecosystem service weighting weighs the functions according to their relation to the services – e.g. the biodiversity variables are 1/3 of 20% - is that right? Please clarify in main text as its needed to understand the results

Response:

- Correct, we now prominently highlighted this, e.g. in the definition box (line 77).

Figure 2 and results: a bit of talking would be good here as help show which of these results are more meaningful and what to take from each. Similarly, clearer communication of the scenarios/weightings and what they represent earlier in the paper, and clearer articulation of the research question would help highlight the most important messages.

Response:

- We now clarified at an earlier text passage the weighting scenarios used and how the weightings were obtained.
- We clarified what are the main effects of interest shown in Figure 2 in the respective section of the Results.
- We clarified in the same section, which ecosystem functions are main drivers for those effects (yield, and flower cover) (line 599-604)

- For better interpretation of the effects of land use and climate type, we added a tukey post hoc analysis (see supplementary material).

P10 – would be interesting to see how sensitive the economic valuation is to changes in prices etc (carbon value seems very high!). How much of this value is market and non-market? What's the market value multifunctionality?

Response:

- We added a sensitivity check to the Results section by using price alternatives, and discussed the results in the Discussion section (see e.g., lines 199-200, lines 690-694).
 - For CO2 price:
 - EU ETS market price for carbon emission (~90 € / t in 2023)
 - Social cost according to Kikstra et al. (2021) of 280 € / t
 - Nitrogen leaching price:
 - For other areas, geographical conditions might cause leaching of nitrogen into groundwater, rather into surface water bodies, causing a social cost of around 3.8 times smaller (1.9 € / kg instead of 7.3 € / kg)
 - Yield price: 40% increase in crop prices (corresponding to real price increases for crops, e.g., 32% for grains in Germany in 2021/22 compared with 2018/19, Bundesinformationszentrum Landwirtschaft, 2022)
- We clarified throughout the manuscript, which is the market value (Crop yield / CO2 using EU ETS accounting price) and non-market value (CO2 price according to Federal German Environmental Agency & according to Kikstra; social cost of nitrogen leaching; biodiversity conservation, soil health) (e.g., lines 369-374, lines 1344-1349)

P10- I've not dug into the methods yet but is it right to compare total value to rental rates? The agricultural yield values are presumably that of the annual crop/fodder, but the carbon cost is a one off- the field doesn't store that amount every year, so the comparison is somewhat misleading. (reading on I now see It is a flux rate- make clear that all the values are some sort of annual yield here)

Response:

- We added a paragraph (lines 409-413) explaining why land rent is a meaningful comparison (as land rent usually represents the economic productivity of the field).
- We clarified that all values are annual values (see figures' captions).

Figure 3 is a cool graph, but the farmer seems to be getting money for regulating services in their value- is that really happening?

Response:

- We now clarified at an earlier text passage (e.g., very prominently in the definition box line 77) which values accrue to the farmers. In particular, higher biodiversity lowers the exposure to risk, and we derive a corresponding economic value (a willingness to pay) for a risk averse farmer (using a standard model of risk aversion from the literature).

Figure 5. Steepest, and positive slopes around t50 is something of a statistical null expectation when using threshold multifunctionality measures- see van der Plas 2016 Nat Comms and Gamfeldt and Roger 2017 NEE, though these two papers differ somewhat in their explanation of such effects This should be acknowledged, or better, shown how much it deviates from the null expectation.

Response:

- We dropped the threshold approach as proposed by reviewer 2 and 3, as it is not necessary to investigate biodiversity - EMF relationships. Instead, we used a linear regression model to investigate the soil biodiversity-egEMF correlation.

P14- it would be good to see which functions/services are driving the decline in EMF under climate change, this should be highlighted and discussed. Some key points from the appendix could move up to illustrate this.

Response:

- We highlight the two main driving ecosystem services 'yield' and 'landscape aesthetics' which are both very much dependent on water availability, and thus are heavily affected by CC (lines 598-604).

P15- Not always clear in the discussion if it is the economic or ecological multifunctionality that is being discussed (or both)

Response:

- We added a definition box (egEMF / enEMF) and changed the naming throughout the text (line 77).

P16- I now see that this is a disturbed soil that is gaining C- this won't be the case for many pastures and meadows that are much closer to equilibrium- this is acknowledged but in a wooly way and this whole section is rather jumbled, vague and hard to understand. Please rewrite to make more specific points.

Response:

- We rewrote the respective paragraph.

P17- the closing paragraph makes sense but says nothing particularly new- i.e. much work has already shown that non-market values need to be considered and included if nature is to be protected. Further, this is already reflected in part in for example, agri-environment schemes that effectively pay farmers to protect non-market values of farmland. What would be more interesting is to see how much more the farmer would need to be paid, and for what, in order for them to make the socially beneficial decision. Also, what would the yield cost be to de-intensify and where would the extra food come from? How sensitive are these options to climate change? The discussion, and paper in general, would greatly benefit from going deeper on

these points as the conclusions are rather generic at present, and this may make for a stronger framing of the paper.

Response:

- We included a discussion of optimal land use type composition, on how land use types will differently affected by climate change and by that, should be in promoted or not promoted considering future climate, and on the role of subsidy schemes for compensating farmers for yield losses when choosing societal beneficial land use types (see discussion section). Hereby, we acknowledge the limits of EMF approaches as shown in this study, as they are lacking information on the expenses farmers have for the management of different land use types. See e.g., lines 803-804)

I was also missing a discussion of what the two approaches showed us that was different, or the same, and what additional insights using both approaches give us. Mixed multifunctionality approaches- that combine ecological and economic measures (or multiple values in the same measure) are also starting to appear- e.g. Neyret et al 2023 Nat Sust. – how does the current approach relate to these?

Response:

- We added a paragraph to the Introduction showing the need for combining enEMF / egEMF approaches (lines 68-75):

“The consideration of both ecological and economic ecosystem multifunctionality indices is a new concept that overcomes the shortcomings of a purely economic evaluation of ecosystems, which often fails to capture the importance of different aspects of ecosystems, e.g., of cultural ecosystem services (Gunton et al., 2017; Daniel et al., 2012). Contrary, ecological ecosystem multifunctionality measures can assess ecosystem services based on their relative importance to different stakeholder groups, which allows to consider aspects, such as overall community-level egENF, and egEMF equity between different stakeholder groups (Neyret et al., 2023).”

- Further, in the Discussion we put more emphasis on the fact that those shortcomings (e.g., the impossibility to grasp the cultural ecosystem service assessed with the economic EMF metric) could be addressed using the mixed approach (lines 747, following).

This paper, as well as Neyret et al 2021 also combine different land uses to see how multifunctionality manifests at larger scales. Such papers highlight the limits of what we can say about identifying the best way of achieving local multifunctionality at small scales- is there some mix of the proportions of the land uses given here that would be the ‘best’ for all? I acknowledge it’s a different question to move to this scale but this point may be worth exploring or discussing in some way.

Response:

- We added a simulation calculating egEMF_Community (mean egEMF scores of all stakeholders) for a set of 1000 different land use type compositions to investigate the land use type composition that provides the highest egEMF for all stakeholders, and to investigate how this optimal composition changes under future climate (see method / results section). Further, we added a simulation to investigate, which land use type composition shows the highest egEMF equity for all stakeholders considered, calculated as the Gini coefficient according to Neyret et al. (2023), and discussed the results.

Methods

P18 'representing all forms of vegetable life'- strange line and probably not true unless it contains the full range of grasses, herbs, lichens, mosses etc.

Response:

- We changed the sentence: "representing a wide range of plant functional types typically found in grasslands in Germany"

p19- justification of microbial biomass as a water measure could be clearer

Response:

- We clarified the relationship: "microbial biomass that can be used as a proxy for the soil water holding capacity (as microbes depend on an aquatic environment to survive and proliferate, and are positively affected by a higher soil moisture)".

p19- which three enzymes

Response:

- Clarified: "three enzymes (cellulase, N-acetylglucosaminidase, acid-phosphatase) that are ubiquitous in most soil microorganisms and represent the carbon, nitrogen, and phosphorus cycles were measured"

p20- deprivation? (you mean concentration?)

Response:

- No, we indeed are talking about deprivation (the removal of nutrients from the soil considering the nitrogen concentration in the harvested biomass). We clarified the sentence: "Mineral nitrogen deprivation (the removal of nutrients from the soil, in kg N / ha) was measured over the years 2016 to 2019 through an elemental analysis of the harvested plant biomass".

p19-20- the methods are a bit too brief here and need to be expanded- sometimes even the basic name of the method is not given, or in the case of the soil biodiversity the taxa measured and how- to what level of taxonomic resolution where the soil organisms measured and how?

Response:

- We expanded the method section accordingly.

p20- the nitrogen surplus method is somewhat coarse with plenty of room for error in almost every part of the equation. The uncertainty should be acknowledged.

Response:

- We amended the respective section 5.3 accordingly to acknowledge the uncertainty and added an alternative price scenario for nitrogen leaching.

P20 – multidiversity- please say which groups went into this measure- plants too?

Response:

- Three different organism groups went into this measure: soil nematodes, soil mesofauna, and soil macrofauna. We now mention the three organism groups in the respective paragraph, and the focus on soil biodiversity is clarified (in the respective paragraph, and throughout the whole manuscript).

P21- its still a mystery as to what the weightings were, and where they came from- the detail can be in the SI but the basics should be here

Response:

- We now made it clear throughout the manuscript that the weights represent stated preferences derived from survey data.

P22- it would be nice to see some sensitivity check of the results to these values- the prices are always going to be somewhat arbitrary, and its also interesting to see how a valuation change can alter the outcome of what the 'best' option is, especially as these prices fluctuate over time in reality.

Response:

- We added a sensitivity check to the Results section by using price alternatives, and discussed the results in the Discussion section. Notably, this approach confirmed our conclusions and strengthened our paper.
 - For CO₂ price:
 - EU ETS market price for carbon emission (~90 € / t in 2023)
 - Social cost according to Kikstra et al. (2021) of 280 € / t
 - Nitrogen leaching price:
 - For other areas, geographical conditions might cause leaching of nitrogen into groundwater, rather into surface water bodies, causing a social cost around 3.8 times smaller (1.9 € / kg instead of 7.3 € / kg)
 - Yield price: 40% increase in crop prices (corresponding to real price increases for crops, e.g., 32% for grains in Germany in 2021/22 compared with 2018/19, Bundesinformationszentrum Landwirtschaft, 2022)
- We clarified throughout the manuscript, which is the market value (Crop yield / CO₂ using EU ETS accounting price) and non-market value (CO₂ price according to Federal German Environmental Agency & according to Kikstra; social cost of nitrogen leaching;

biodiversity conservation, soil health).

8.1 There is some important info here that would help make sense of the more synthetic measures- I would suggest saying which were the main responses/variables driving summary measures a bit more in the main paper to help guide the reader through what is going on.

Response:

- We added a paragraph in the Results and Discussion section elaborating the main ecosystem functions / services that drive EMF - especially the decrease of egEMF under future climate (the two services 'yield' and 'landscape aesthetics' which are both strongly dependent on water availability, and, by that, heavily affected by changed precipitation patterns (e.g., lines 598-604).

Figure 7. Real data on stakeholder priorities in Germany show a far more even spread of priorities than is seen here- they tend to acknowledge each other's needs a lot more than you might expect (see Peter et al 2022 People and Nature). Some discussion of this or adjustment of the weightings in a sensitivity analysis if going deeper would be good. The fact that these are idealized, researcher defined priorities, should also be acknowledged in the main paper.

Response:

- We replaced the estimated weightings with weightings based on measured stakeholder priorities in Germany (Peter et al., 2022).

Responses to Reviewer #2:

The manuscript explores how land-use type and climate change affect ecological and economic multifunctionality using a 7-year manipulated field experiment simulating five common land-use types (three grassland and two farmland types either sustainably or intensively managed) and future climate in a site. The authors concluded that sustainable land management enhances ecological and economic multifunctionality under both ambient and future climate. Overall, this work was of important value considering that its experimental design can help us simultaneously examine the effects of two major drivers--land use and future climate change on multiple ecosystem service and functions, which is less considered in previous studies. (I cannot give specific detailed comments due to lack of line number in the whole ms)

Response:

- We acknowledge the positive and constructive feedback on our work!

My major comments and concerns about this ms:

1. I don't think novel of this study lies in comparing sustainable land management with intensive management effects on EMF, because it is widely accepted that sustainable land management can offer more ecological service functions than intensive land use which only consider short-term high production functions. Instead, the novel of this work should focus on the interaction of land use and future climate, I therefore more wonder which land use will be confronted with great risk in offering ecological service functions in future climate changes. Unfortunately, the author showed that the interaction effects of LUT and climate was non-significant. I noted these results are from overall analyses including all five land use types (table 2, 3). However, this five land use types include two categories: grassland and farmland, and each included sustainable and intensive management. In your statistical models, whether these factors were considered? Grassland is greatly different from farmland, whether you attempted to conduct these interaction analyses in grassland and farmland, respectively, when you compare sustainable and intensive land management in future climate change. In your current results I cannot find these analyses. Moreover, whether you analyze the interactive effects of climate change and land use based on provisioning service, regulating service, and supporting service, respectively? Maybe these analyses could obtain new important findings.

Response:

- We conducted a statistical analysis considering the different land use categories that was not included in the manuscript before, as no significant interaction effect was shown. Still, through the inclusion of an additional ecosystem function (flower cover) approximating an additional ecosystem service (landscape aesthetics), an interaction effect was shown for different weighting scenarios (according to preferences of two stakeholders) which allowed conclusions regarding the varying capacity of different land use types to mediate the effects of future climate on ecological ecosystem multifunctionality.

2. For this work, another important and novel point was the considered economic measure of EMF value, especially for those economic value of ecological service provision excluding food production, which is less evaluated in previous EMF research. Future land use not only should consider its production value also include its economic cost or value from the other ecological service provision. So, this work is of important value to help guiding government to evaluate land use, also farmer to trade off how to use land more economically. I therefore suggest author more highlight this point in main finding and conclusion.

Response:

- We put more emphasis on the economic EMF index, on the strong correlation between the economic EMF index and the ecological EMF index, and on the shortcomings of the economic approach that makes it necessary to use mixed ecological-economic multifunctionality metrics (e.g., lines 68-75). In the Discussion, we highlight the benefits that the economic approach can bring, whereas the ecological approach shows indifferently high egEMF provision for different land use types (e.g., lines 747, following). We emphasise the need to consider the promotion of sustainably-managed grassland which is less affected by climate change and has an equally high egEMF provision for 2 / 4 stakeholders compared to sustainably managed farmland, but a substantially higher economic EMF provision.

3. This study also analyzed the biodiversity-EMF relationship. However, I don't think these analyses are necessary in this work. Moreover, in this biodiversity-EMF relationship analysis,

plant diversity seems not be included in this multidiversity index (I was not completely sure for this, because many data analyses method are not clearly introduced in this ms), while role of plant diversity in maintaining EMF cannot be neglected. I also wonder whether plant diversity was included in your ecological and economical multifunctionality? Because plant diversity conservation in land use should also be very important. According to Fig1, this seems not be included. Again, why do you use different EMF approach in analyzing biodiversity-EMF and land use/climate effects on EMF, one for threshold approach, the other for averaging approach. Both approaches should be examined in all analyses to solid these results. Moreover, I also noted that all functions except yield included in EMF index are about belowground soil functioning, so the current EMF index was more represent for soil EMF.

Response:

- Indeed, we only focus on soil biodiversity in our analysis, as it was shown that soil biodiversity is a major driver of EMF (Delgado-Baquerizo et al., 2020; Schuldt et al., 2018; Delgado-Baquerizo et al., 2016; Soliveres et al., 2016; Wagg et al., 2014). We made this clear throughout the manuscript (e.g., Figure 1, Section 5.4).
- We dropped the threshold approach, as proposed by reviewers 2 & 3, as it was not necessary to analyse the biodiversity - EMF relationship.

The other comments:

1. I am missing some important details on the methods. For example, although the authors showed that there were 50 plots, and each approximately 400m². How many replications for each treatment? And for the extensive pasture (EP) that includes grazing with sheep (~20 sheep grazing 24 hours per plot) which takes place three times a year (early spring, mid- to late spring, and mid of summer), are the large steel constructions for climate manipulation is removed in the grazing days? Also, for the samplings and measurements, all the samplings were conducted from 2014-2020, and you have the 7-year data? All the plots were harvested, or there have some sampling quadrats? How many sampling you have in total? There was no sampling replications? In addition, I noticed that you have mention the subplots and subsamples, what are these mean? soil sampling depth? These need clarification. In general, the quality of the current paper can only be fully assessed once these details are fully understood.

Response:

- We apologise for the lack of information/detail.
- There are five replicates for each land use type-climate type combination; this is now clearly highlighted in the Introduction and in each figure's caption.
- Steel constructions are put in concrete and are permanent (we added the fact that they are permanent, line 857).
- Not for all 14 different ecosystem functions, data was available for every year. We added an overview table showing which year/data was used for which ecosystem function (see supplementary material, table 17) and clarified this within the text
- For the four land use types, EM, IM, OF, CF, the whole plots were harvested with a combine harvester. For EP, which was grazed with sheep, this was not possible. To assess biomass yields, 4 sheep-excluding subsample plots were manually harvested right above the soil, and biomass within those four plots was compared with biomass within four subsample plots that were grazed by sheep (see section 5.2).

- We clarified the wording ('subplot' was removed, as it was misleading and should refer to the 50 plots of the experiment).
- There was sampling replication, differing for different ecosystem functions that were assessed. We expanded the Methods section to explain all samplings in more detail, including the soil sample depth.

2.I also concern about the data analysis in this manuscript. The statistical analysis section is very brief currently, as the authors only provided the R packages for the analysis. I would also like to see more details on the actual statistical analyses:

- Do you use the multi-year data, how do you treat these data?
 - Measurements were indeed taken across several years (see table 17 Supplementary material for an overview). However, as we didn't have specific hypotheses about temporal changes in response variables, we used the arithmetic mean across years for each response variable rather than applying ordinary repeated measurements. This is in line with many previous ecosystem multifunctionality assessments (e.g., Meyer et al. 2018), and we clarified this throughout the manuscript.
- Only use the mean, or conducted the repeated-measure ANOVA or the linear mixed effects models?
 - For statistical analyses, we applied linear mixed effects models to plot-level data, with the factors climate and land-use as well as their interaction as fixed effects. According to the split-plot design of the experiment, Mainplot nested in climate was included as a random effect (R automatically detects the nesting of Mainplot in Climate) and used as the error term for the effect of climate. For statistical testing, we used F tests based on type III ANOVA and Satterthwaite's approximation. See lines 1406 - 1432.
- Also, it seems that the analysis about the interactive effects of land-use type and climate on EMF is using the Type III ANOVA Satterthwaite's method in accordance with the split-plot design of the experiment? Do this analysis using the plot level data
 - Yes, was now made clear in line 1413.
- Do they include random terms?
 - Yes, clarified (in section 'statistical analysis').
- Based on the figure 2, 4, dots indicate the multifunctionality level within the plots of the experiment, triangles indicate group means, it seems there are many plots in each treatment. What about the sample size?
 - Sample size was added to all figure captions.
- Why table 2, 3 showed that the interactive effects of land-use type and climate on EMF was analysed by general linear regression model
 - This was a typo, the general linear regression model table is actually the linear mixed effects regression model table → corrected.
- So, how do the used models differ from the ANOVA analysis?
 - See above.
- Moreover, the authors have not provided the method for the biodiversity–multifunctionality relationship analysis. Regression analysis? Person correlations? Linear mixed–effects models?

- For this analysis, data in all the land-use types are included?
 - Yes, → was made clear, e.g., in line 729, we used a general linear regression.
- And it was analysed in the ambient and future climate, respectively
 - Yes, it was analysed separately for both ambient and future climate; we clarified this, e.g., in line 494 .
- Again, this does not mean that the analyses are wrong, but more details on the analyses should be provided.
 - We clarified this throughout the manuscript and added a more detailed explanation of the statistical analysis in section 5.9 - statistical analysis.

3. I cannot understand why author analyzed EMF for all functions and also for all services In Fig 2? what's the calculating difference between EMF equal weighting of all functions (a) and all services (b)? According to fig 1, ecosystem services are also from these functions, it seems that they are from the same set of functions data.

Response:

- We use the equal ecosystem function weighting and equal ecosystem service weighting to align to earlier research. Still, in the revised version of the manuscript, we shifted those two weighting scenarios to the annex, and used weighting according to two additional stakeholder preferences instead, based on measured stakeholder preferences from survey data (Peter et al., 2021), as inspired by reviewer 1 (see also responses to reviewer 1 above).
- Some functions consist of different services (e.g., 6 services for soil health → to avoid overweighting of this service, the 6 functions were weighted with 1/6 weight. We made this clearer throughout the manuscript (e.g., line 77, Definition box).

4. The manuscript still needs a few improvements in terms of content and structure, especially for the introduction section. As the introduction nicely starts with ecosystem multifunctionality (EMF), however, the EMF has been widely investigated, it would be nice if they could point out the importance of considering both the ecological and economic multifunctionality. Not only show that here we do this. Then the authors introduced the land use, climate change, as well as the biodiversity as the drivers of EMF, however, all these factors included in the second paragraph, reads confusing and loss strength. It also leaves a reader wondering about what is the real novelty and significance which is relevant for publication in this journal. All the sections are not well-connected currently. These contents in the introduction need to be logically re-organized. Another key point is about the experimental duration, the authors should emphasize the multi-year data, which is important metrics of this study, as many studies using the one-year data, there may be the time-lag responses of ecosystem to land-use and climate change. In addition, I think that the discussion section could be shortened and improved. The discussions in each paragraph also are not well-connected currently.

Response:

- We strengthened the storyline and readability throughout the manuscript, starting with an explanation of the need to combine economic and ecological approaches to overcome shortcomings of the single approaches (Introduction, e.g., lines 68-75), and in the Discussion we put more emphasis on the fact that those shortcomings (e.g., the impossibility to grasp the cultural ecosystem service assessed with the economic EMF metric) could be solved using the mixed approach, see e.g., lines 747, following).

- We clarified the use of the multi-level data by using averages of all ecosystem functions over multiple years (see tabular overview, supplementary material)

Lastly, there was not line number in whole ms, it's very trouble when comment some specific points, so I strongly suggest adding line number in new revision ms.

Response:

- We apologise and added line numbers in the revised version.

Responses to Reviewer #3:

In the manuscript « Sustainable land management enhances ecological and economic multifunctionality under ambient and future climate” the authors investigate the impact of a crossed land use and climate change experiment on a range of ecosystem multifunctionality metrics, focusing on five main services: food production, climate regulation, water regulation, soil health and biodiversity conservation. They also propose an economic approach to multifunctionality to assess the economic impacts of said changes on society at large. They show that in general, MF would decrease in future climates, especially when considering farmers’ priorities, and less so when all services are considered equally (equal weight or environmentalists’ preferences).

This study addresses an important question for agro-ecosystems management and functioning in face of climate change, and the analyses seem sound. However, it would benefit from a tighter framing of the research questions and their corresponding metrics and analyses to address. As it is, some analyses are not fully justified or introduced. There are also a number of limitations regarding the calculation of multifunctionality that should be acknowledged.

Response:

- We appreciate the positive and constructive feedback!

Main comments

1. The authors provide a range of analyses, assessing how multiple multifunctionality metrics (7 in total) respond to climate and land use. However, **why exactly each metric or analysis was selected is unclear**. For instance, it might be unnecessary to show both the equal weighting of function AND the equal weighting of services, which adds to the overall complexity, unless more justification is provided on why each of these metrics is important. The other analysis could be moved to the supplementary material.

Response:

- The equal function and equal service weighting was used to align with earlier research (e.g., Allan et al. 2015) . Still, those two weighting scenarios were shifted to the annex in the revised version of the manuscript. Instead, in the revised version of the manuscript, only real stakeholder stated preferences data based on survey data was used (instead of estimated preferences for only two stakeholders), as inspired by reviewer 1.

Besides, the threshold analysis comes a bit as a surprise at the end, and is currently a bit disconnected from the rest of the paper (e.g., why test climate and not land use if the rest of the paper tackles both issues?). The topic on climate impact on BEF relationship is introduced in the introduction, but explicitly stating what you're going to test (e.g. does climate change affects the strength of the relationship?) might help.

Response:

- We dropped the threshold approach as proposed by 2 reviewers, as it was not necessary to analyse the soil biodiversity - egEMF relationship, which instead was analysed with a linear regression model under both ambient and future climate.

2. There are also some limitations that should be acknowledged, or clarified, regarding the calculation of multifunctionality.

- Services used: how did the authors choose the services? Do they really represent the most prioritised services required by local stakeholders? Would the inclusion of other services (e.g. cultural services) affect the results?

Response

- We acknowledge that the chosen set of services is incomplete (see e.g., lines 550-555).
- We included an additional ecosystem function (flower cover) approximating an additional ecosystem service (landscape aesthetics) to represent a more diversified set of ecosystem services that is consistent with other studies choosing sets of ecosystem services (Allan et al., 2015). We discuss the somehow arbitrary choice of ecosystem services in the Discussion section.

- Besides, it should be made clearer that the priority scores are made-up scenarios, not based on any social-ecological data (here the justification is limited to a citation of Manning et al 2018, but the scores used are different anyways). This, together with the fact that essentially only two stakeholder scenarios are considered (farmers and environmentalists) calls for

Response

- This comment seems to be cut off, but we think we got the point and address this comment by now using real stakeholder data based on a published survey instead, and included four instead of two stakeholders (farmers, locals, environmental conservation agencies, tourism sector)..

• Biodiversity: from what I understood, the biodiversity measures include essentially soil fauna. This is alright for using biodiversity as a function, or as a driver of multifunctionality. But I'm not sure it works as a service, because when people value biodiversity protection they rarely think about soil fauna, but rather plants, birds, etc. I suspect not much can be done about that in the manuscript, except acknowledging it.

Response:

- We acknowledge the fact that soil biodiversity is not the most intuitive part of biodiversity that is valued by people. We still include soil biodiversity, to reflect that people appreciate soil biodiversity for its beauty and out of curiosity (Philips et al. 2020), see e.g., line 147.

• Soil health/food production: usually, ecosystem services multifunctionality (as in Manning et al. 2018) only includes final benefits, ie the benefits that directly benefit people, but not regulating services. This is to avoid double counting (e.g. pollination and the boost in food production provided by pollinators). There might be a similar problem with soil health here, which underpins food production and water regulation. Is there a risk that this would affect the results?

Response:

- Considering the complexity and the inherent interlinkage between different ecosystem functions and services, it is true that we have to be careful to avoid double counting. Thus, we did not include the effect that soil health preservation is increasing expected yield. We did, however, include the yield-stabilising effect of soil health, as the associated insurance value comes on top. We have explained this more carefully now in the manuscript (see e.g., lines 1367-1369).

As a smaller comment, the terminology would benefit from being simplified, along with homogenised acronyms (if they are really needed). For instance, "economic multifunctionality" could be called "overall/total economic value" to homogenise it with the "farmers' economic value", which would remove the need to write "ecological ecosystem multifunctionality" for the other metric. Some acronyms are also only used towards the end of the manuscript (FEV, eEMF) – are they really needed? If so it would be good to introduce them earlier on and use them consistently.

Response:

- We updated our terminology and added a definition box (egEMF - ecological ecosystem multifunctionality / enEMF - economic ecosystem multifunctionality) and updated the terminology throughout the whole text (e.g., line 77).
- We refrained from using the term "total economic value", as not all ecosystem services could be assessed or monetised, so "total" would be incorrect.

I also have several minor comments on the manuscript. As the pdf I received did not have line numbers, I'm attaching a commented pdf instead.

Comments from pdf:

Abstract: "Here we assess how multifunctionality of agroecosystems in Central Germany depends on land-use type and climate change." → Proposal: "depends on land use type and climate"

- Was changed accordingly.

Abstract: "Above-belowground biodiversity" → The references in the methods (Yin et al 2019, Siebert et al. 2020) only mention below-ground biodiversity.

- We adapted this sentence ("Soil biodiversity" instead of "above-belowground biodiversity") as all the biodiversity datasets used for this study considered soil biodiversity (soil mesofauna diversity, soil macrofauna diversity, soil nematode diversity).

P 3 Introduction: This is a matter of preference, but I would suggest starting with a broad topic / larger-scale issues rather than definitions

- We adapted the Introduction accordingly.

P4 "Despite the field of EMF research gaining more and more attention (Giling et al., 2019), the impact of different land-use types and future climate on EMF in the context of agriculture with a direct quantification of those changes in monetary units has not been assessed before." → Maybe add one sentence here saying why it's important for more global issues (not only it's not been done)

- We explained the need to combine economic and ecological approaches to overcome shortcomings of the single approaches (Introduction, e.g., lines 68, following), and in the discussion we put more emphasis on the fact that those shortcomings (e.g., the impossibility to grasp the cultural ecosystem service assessed with the economic EMF metric) could be solved using the mixed approach (see e.g., lines 747, following).

"This study aims to address this research gap." → Here it might be worth making your other question (will climate change modify the strength of the BEF relationship) explicit as well

- In the revision of the manuscript, we put less emphasis on the question if climate change will modify the strength of the biodiversity-egEMF relationship, but we strengthened our research question and narrative around the future climate effect and the ability of different land use types to mediate the effects of future climate. This was clarified throughout the text.

P 5: "microbial biomass, enzymatic activity, and decomposition rate indicate soil health preservation" → Is this service underpinning water pollution control/food production? is there a risk of double counting in the MF measure?

- See our response regarding double counting above.

“Biodiversity of soil nematodes and meso- / macrofauna indicates biodiversity conservation” → These groups underpin ecosystem functioning so it makes sense as a "function". But it might be worth mentioning as a limitation that most people would not think of soil fauna when asked which part of biodiversity they value (i.e., plants, birds, etc) so it might not be the best indicators for this service

- See our response on soil biodiversity appreciation above.

P 6: Nice figure (Figure 1)! the "ecological approach" vs "economic approach" are a bit confusing though, as sometimes you define the ecological MF based on services (which link to economic MF here)

- We removed this part from the figure and clarified it in the figure caption.

P 6: “The weighting was done based on different objectives and stakeholder preferences” → "scenarios of stakeholder preferences" maybe?

- We adapted this sentence: “The weighting was done based on preferences of four different stakeholders, derived from published survey data (Peter et al. (2021), see Figure 7, Supplementary material).”

P 7: “Further, we tested if certain ecosystem functions drive EMF.” → Do you mean biodiversity? not clear otherwise

- Yes, the aim was to test if soil biodiversity drives EMF as shown in numerous different studies (Hong et al., 2022; Eisenhauer et al., 2018; Delgado-Baquerizo et al., 2017; Soliveres et al., 2016; Allan et al., 2015; Wagg et al., 2014). We clarified this in the respective paragraph and throughout the whole manuscript.

P 7: “Further, we tested if certain ecosystem functions drive EMF. To analyse biodiversity-EMF relationships, we assessed EMF based on the multiple threshold approach” → The justification of why this approach is needed (on top of the average approach) is missing

- We dropped the threshold approach as proposed by 2 out of 3 reviewers, as it was not necessary to analyse soil biodiversity-egEMF relationships (which was done using a linear regression model instead).

P 7, section 3.1: This part is a bit dense due to the number of MF metrics used

- See below (we clarified the different EMF metrics using a definition box (see p) and consistent naming throughout the manuscript).

Figure 2 - How many plots per treatment? please provide sampling numbers in the figures. Besides, not all dots are visible in the figure - maybe some jitter or transparency would help

- The design includes five replicates for each LUT:Climate combination: This information was added to all figure captions.

Figure 2 - Just to make sure - does that mean no interaction was significant?

- This was true for the old version of the manuscript, correct. However, the inclusion of an additional ecosystem service (landscape aesthetics) and the adaptation of the weighting scenarios based on measured instead of estimated stakeholder preferences based on the reviewers' comments revealed land use type - climate interaction effects for two out of four weighting scenarios.

P 9: "For sustainably-managed grassland, economic EMF is around twice as high as the monetary ecosystem service provision for farmers" → It would be easier to follow if similar terms were used, e.g. overall economic value and farmer's economic value or equivalent

- We improved the consistency of the terms used throughout the manuscript (see e.g., definition box, line 77).

P 10: "For intensively-managed farmland, monetary ecosystem service provision for farmers is 1.42 times higher than the total economic EMF value" → Say why: economic "loss" due to water pollution, which is not considered by farmers

- Clarified (see e.g., lines 358 - 359).

Figure 3 - eEMF not introduced in the text (and not used in the figure, maybe not needed?)

- We improved the consistency of the terms used throughout the manuscript (see definition box).

P 13: "No significant difference was found regarding the strength of the biodiversity-EMF relationship under ambient or future climate (Tmin at threshold $t = 29\%$ for both ambient and future climate, Tmax at $t = 71\%$ for both ambient and future climate, Tmde at $t_{amb} = 51\%$ / $t_{fut} = 49\%$)" → What exactly does the comparison of Tmin/Tmax brings (eg what research question does it address?) The strength at Tmax seems different for the ambient vs. future climate, maybe worth mentioning

- We dropped the threshold approach as proposed by 2 out of 3 reviewers, as it was not necessary to analyse soil biodiversity-egEMF relationships (which was done using a linear regression model instead).

P 15: "This finding is in line with previous work showing that species-rich grasslands reduce nitrogen leaching from soils" → In this specific case isn't the explanation more likely to be no fertilisation (rather than more species) in extensive grasslands?

- True, we weakened this statement and added that this is probably mainly due to fertilisation (see e.g., lines 649 - 650).

P 17: "Agricultural land is typically managed by farmers. However, those are the only stakeholder group considered in this study" → Only two stakeholder preference scenarios are included so this can't be generalised too much

- Adapted, as we now have 4 stakeholder groups considered.

P 21, Headline 5.6 - If you the distinction ecological/economical MF is kept, the title should mention 'ecological EMF'

- Adapted.

P 31 - Supplementary material → It would be worth adding one figure showing the tradeoffs between services (eg correlation matrix)

- We added a correlation matrix showing correlations for all ecosystem functions, services, and EMF (egEMF & enEMF) (see Table 16, Supplementary material).

Reviewers' Comments:

Reviewer #1:

Remarks to the Author:

The authors have significantly revised the paper and put considerable effort into addressing my concerns. I appreciate this. The new analyses, in which new multifunctionality metrics are presented have given greater insight and relevance to the results (but additional caveats should be added to the paper surrounding their presentation and interpretation). However, this was a frustrating paper to re-review as many aspects of the paper are not fully worked up or presented to a sufficient standard.

In general, several sections need re-writing, both at a within sentence level but also in terms of overall structure and content, to improve the logical flow of arguments and the presentation of ideas. These are detailed below but in particular the introduction, the ecological multifunctionality part of the results and the mid-discussion need attention.

While the new analysis is improved and does provide more insight, it is not properly justified or explained, or properly compared to the economic approach. Also, the landscape simulations are a nice addition but not clearly presented or interpreted. As the data is taken from 1 experiment/field, it's perhaps a bit much to extrapolate these findings to landscape scales. The site in question will show only a small range of possible values for the different land use types, and so this part needs to be treated with caution. I actually feel it might be easier to omit it altogether.

I also have some concerns about how the Peter data was used. The services surveyed there were 'final benefits' (sensu cascade model), but most of what is measured here is regulating/supporting level (depending on which ES classification scheme is used). The authors reallocate or guess at stakeholder priorities to the point I feel this data can only be said to loosely resemble the original survey, and the measures to represent what stakeholders said they prioritised. Wording should be toned down and caveats added to acknowledge this.

A small point, but I also suggest calling the measures ecological or stakeholder based and or monetary or economic multifunctionality, and writing these in full. It would take up a few words but make the paper much easier to read- I constantly had to keep reminding myself which MF measure I was reading about.

Below are more detailed points which I hope can be useful in revising the paper.

Specific points:

Throughout- replace 'farmland' with cropland or arable land (ackerland), as farmland often means any agricultural land including that used for livestock.

25-28 intro ok here, but there isn't really a 'research gap' statement.

26. Omit functions or revise to account for these as underpinning services

36. not clear what ecological multifunctionality is here- one of the stakeholder weighted measures or all of them?

37 according > based upon...does intensively....show higher...when compared to

43-45- conclusion still feels a bit generic and not really closely tied to results. E.g. you now know that the current prevailing land management set up suits the farmer (which is why they manage as they do) but not the rest of society, so it indicates that the rest of society needs to incentivize/pay the farmer to get the outcome they want/need for sustainability.

58. this point was better covered by Byrnes et al 2014 than Manning 2018.

58-60. more that this approach is more meaningful if working at the level of ecosystem services

64- would be good if the intro before getting here stated that: 1. we need ways of measuring overall social impact of global change on overall performance of ecosystems and that 2. Economic valuation and multifunctionality measures are a way of doing this but there are pros and cons to different approaches...

68- allows consideration of other aspects such

70 – trade-offs between stakeholders a key aspect here: see also Linders et al 2021 People and Nature.

72- 'can be used to approximate'- more accurately 'can be used as indicators'.

Stakeholders>Stakeholder groups

80- see Le Provost et al 2023, NEE for a more up to date reference on land-use-biodiversity-EMF relations (Hector and Bagchi did not look in an agricultural context from what I remember)

80 Here > However, at the level of ecosystem services (and move sentence 82-84 before this one)

90- found>received

90-94. Poorly written- please revise

100- see also <https://www.nature.com/articles/ncomms9159> and the many papers by Fernando Maestre and others looking at climate MF relationships

102- Fischer et al., 2010 does not look at multifunctionality from what I can recall

107-111 – the gap is not expressed clearly here but it needs to be. There are lots of papers on how changes to climate and land use will affect economic properties – IPCC and IPBES reports are full of them, so do you mean how these drivers affect EMF measures that include economic properties? Neyret et al 2023 had some economic properties mixed with ecological ones, but done coarsely, and I can think of another Gras et al, that trades off economic and ecological multifunctionality, Grass et al 2020 <https://www.nature.com/articles/s41467-020-15013-5> but there is something different here to those in these papers- please clearly highlight the novelty and need for the approach used.

137-139 As much as I'd like them to, I'm not sure if people really appreciate soil biodiversity- most don't even know it exists! It's only just beginning to be appreciated by professional ecologists, though interest is rising in the popular imagination too, which is great. I would imagine a survey of preferences/valuation of biodiversity would put this group very low. In the figure it seems its classified as supporting anyway (Table 7 confirms this). Still, we could probably debate weightings, ES classification schemes etc. eternally. While I don't fully agree with all the choices made in this aspect, I accept them as they are justified at least. But the divergence of what Peter et al measured and what is presented here needs to be acknowledged (see later comments)

Figure 1. Inessential, but pollution is a human value, so should maybe be in services- a flow of: nitrogen surplus, soil nutrient concentration (as the indicator) and water quality as the service would be more accurate there. Flower production>flower cover may also be more accurate.

164- stakeholder groups

164-165- would be useful to state roughly what the main differences in preferences were here, so results can be understood, and that this data was manipulated considerably.

190- the response letter suggested some landscape simulation was done but I don't see this described in the approach part? (Reading on this either has to be more focal or omitted)

191-211- the results are somewhat listed off here- could do with some explanation or highlighting of major trends and patterns

214- this simulated landscapes part comes in out of the blue- it needs to be in the approach section. The results also need to be presented in a more structured way so that they are more distinct and the messages behind each part are clear. (it might be easier to omit this part)

Figure 2. Stakeholder weightings or contribution of individual ES not really shown anywhere so hard to understand what is behind these trends

Table 2. A lot of abbreviations presented- what are all these? Please write out in full. The values are percentages presumably? Better presented as 4 pie charts perhaps?

Table 3- a main paper item? Perhaps present key values in text and put rest in appendix.

257- these results are also listed off, but should be compared to the other EMF metrics. What changes when different services are introduced or omitted- make it clear and easy for the reader. The whole economic section is written in a different style and has a different feel to the section preceding it. Please harmonize.

268 not not

304- please give a quantify this strong correlation – is it in the realm of 0.4 or 0.99- I've seen both described as strong.

306- this difference sounds interesting- can you expand and say why its different

Figure 3. Really nice figure.

Table 4. see comments on Table 3

336. This section feels a bit of an afterthought, a correlation is seen but would you really expect soil fauna diversity to be driving all these ES? A correlation is found but not much is done to properly separate the effects of diversity from other drivers (all the low soil BD are also arable fields so a totally different system). Options: expand, omit, place strong caveats, or justify more strongly.

361- but a clear comparison was not made in the results

365- please clarify what the differences were

366- this study or any study?
369- representative of what?
387- discussion seems quite climate focused, relative to its prevalence in the main paper
397-398- was the data to support this argument presented?
407-409 Note that the survey of of Peters et al asked stakeholder to weight services at the final benefits level only, and so regulating/supporting services underpinning final benefits are not included. This will likely have influenced your results, which include a lot of these regulating services. This needs to be discussed.
404-419 I found this passage a bit confusing at it seemed to start with a discussion of ecological EMF, then discuss economic- needs revision/clarification.
458. Unclear sentence
483- farmland might be better called arable cropland here and elsewhere as pasture is also farmland
497- new paragraph here
517- certain threshold of what?
526- the landscape composition part is very underdeveloped- expand or omit. The data appears rather shoehorned into this approach and I'm not sure if its really suitable for it, given that its effectively taken from a single field, and represents only a small subset of landscape services.
529- not clear what is meant by linear models here
539 indifferent- do you mean moderate/intermediate or similar? I think the latter
548 onwards- discussion is much stronger here
651- how well does microbial biomass approximate water- there are a lot of other factors influencing microbial biomass so maybe not that great?! (also plant growth, I thought we were talking about microbes?)
826- mean
832- biodiversity conservation in Peter et al was not a supporting service so this is questionable, also values have very much been assumed here, and others are missing. Acknowledge and discuss
844- where do these numbers come from? Peter et al show farmers giving around 15% of their score to biodiversity (and then as a cultural service), much lower than the 29% given here. I guess this is 29% of the parts that there was data for? Given the amount of data manipulation to fit the services measures it might be best to say these weightings are loosely based or inspired by this data as they are quite far removed from them.
856- Neyret et al used aggregated measures of value from land uses when combining land classes – how was this done here?
Table 7. Peter classified biodiversity as a cultural service (enjoyment of wildlife etc).
Table 7. Should the two food scores not be summed rather than averaged as they when combined represent the overall importance of food.
1540 Notes need to be deleted.
Table 16 Is poorly formatted and can't be read

Reviewer #2:

Remarks to the Author:

The authors have accommodated all my comments to my satisfaction. Very interesting and important contribution. Well done.

Reviewer #3:

Remarks to the Author:

Thank you for the opportunity to review this new version of the manuscript. The authors have revised significant portions of the text to improve clarity regarding the protocol, analyses, and multiple metrics used in the study. They integrated new data of "real" ES prioritization from an earlier study, and conducted a whole new set of landscape simulation analyses to identify landscape compositions that maximize their different indices of multifunctionality. I commend them for these in-depth revisions. Unfortunately, one major issue of the previous version of the manuscript (also noted by Reviewer 1) was the lack of a tighter framing and take-home message of the study. This has been addressed but only partially (l. 60+). Further, the new analyses are not

introduced at all and come “out of the blue” in the result section, which is a shame considering the effort I expect went into them. I do think the findings, and new analyses, are of interest but I strongly suggest additional effort, in particular in the introduction, to justify their scientific relevance and highlight research gaps as well as objectives and specific research questions of the study. This would also help highlight a few main take-home messages in the discussion. Please also find some more specific comments below.

Specific comments

L. 39: be more specific: which land-use most mitigates climate change?

New analyses regarding optimal landscape composition are not mentioned in the abstract

I. 65 onwards: I appreciate the justification of both indices of multifunctionality. Part of this could go later in the introduction, after some hypotheses are more clearly introduced, and along with some research questions (see below for I. 115-116). This section is mostly clear about what ecological multifunctionality brings compared to the economical one, but does not say why the economical one is needed.

One additional argument to this part could be that traditional (non-weighted) approaches of multifunctionality do not take into account the trade-offs between the priorities/demands of different people, and thus the possibly contrasted benefits they gain from ecosystems – which both the egEMF and to some extent the enEMF (with the different benefits for farmers and overall) can integrate.

L. 67: I don't understand what “driven by EMF” means here

I. 71 (also 104): contrary -> conversely

I. 74: the equity part comes out of the blue here, it would need some context regarding why equity is considered or what it brings to the topic. I would suggest to just skip equity and focus on multifunctionality (also in the landscape optimization: use only multifunctionality as criteria) or to add a full introduction and discussion of equity in terms of trade-offs between different stakeholders, limiting conflicts, etc + relevant literature (but this would probably lead to too many concepts introduced in this already dense paper, the safest might be to skip it).

L. 77 Definition box -> glossary

I. 82-107: here the text jumps a bit from topic to topic climate – biodiversity – BEF – effect of climate on biodiversity – maybe could be streamlined a bit to better introduce the next paragraph? A bit of reorganization and a few additional words would help to make the arguments (which are already there but more or less implicitly) more explicit (e.g. in the line of “we know climate will affect individual services, but because people value services differently the overall impact on stakeholders well-being and economic benefits remain unknown”; “climate change (combined with anthropogenic impact) affects biodiversity, but as BEF relationships might change too the overall impact on ecosystem functioning is unclear”

I. 115-118: this part should be expanded (maybe as a new paragraph) to justify and introduce all the analyses covered in the paper, including the BEF and landscape simulation analyses. The gap addressed by this study also goes beyond the use of monetary units as this is one among multiple indices used to address the combined effects of land use and climate change on these systems

I. 147: unclear – what is not included is the intrinsic value? But the insurance value is, is that right?

I. 168 onwards: some details (eg. Data exclusion for the first two years, process to average indicators across years, detailed economic quantification, sensitivity analyses) don't need to be repeated here from the methods/results

l. 203: the broader, expected outcome of the paper is currently a bit hidden here following the description of sensitivity analyses – could be moved to elsewhere in the intro or to the discussion

l.234 onwards: very long sentences, difficult to follow.

“The weighting according to preferences of the tourism...” and similar could be shortened to e.g. tourists egEMF here and elsewhere.

l. 252: I suggest removing the equity-related analyses (or fully introduce them in the introduction).

l. 252: long and complex sentence, rewrite?

Fig 2: I might have missed it in the results and discussion, but it seems that the OF treatment, despite having relatively higher EMF than the other land uses, is clearly the one that would have the largest decrease under climate change; Does that mean that OF is actually more sensitive to climate change, or simply has more “room” to decrease as it starts from a high value? Maybe worth discussing

l. 409 onwards: is it still the sensitivity analyses section or a new one?

l. 496: In fig 5 the light blue pvalue (which I expect corresponds to the future climate) says 0.06, not < 0.01 , homogenise (or explain the difference)

Fig 5 what is the black line? Some confidence intervals should be added. Is the interaction biodiversity x climate significant? What is the R² and pvalues in black on top?

Discussion: I found the beginning and end of the discussion interesting and useful, the middle part is a bit harder to follow as it switches back and forth from repeating some results, discussing specific services (eg leaching), or study site specificities or broader land-use change considerations. I don't really have a constructive suggestion on how to improve this, unfortunately, but maybe having separate paragraphs for the different themes and focusing on explicitly answering/discussing the different research questions would help.

l. 1133: maybe I missed it, how do you calculate landscape-level service supply (e.g. average, sum? Do you average biodiversity values or use gamma diversity (it should be gamma diversity)?

l. 1141: technically power is not needed to calculate community-level multifunctionality, and you don't use it anyways, so you can remove this sentence

Report how we have revised the manuscript

Sustainable land management enhances ecological and economic multifunctionality under ambient and future climate

For Nature Communications in response to the reviewers' comments

We are grateful for the constructive comments by the editor and the reviewers. We feel that revising the manuscript and addressing these comments has made it much more convincing. In the following, we explain all changes we have made in response to the review reports. To this end, we reproduce the reviewer comments and provide the respective answers in *blue*. We have added numbering to the report by the second reviewer to facilitate referencing.

Our main revisions have been the following:

1. Acknowledgement of the caveats related to the transfer of stakeholder priorities from Peter et al. (2021).
2. Removal of the analysis of community-level multifunctionality and multifunctionality equity as recommended by reviewers 1 & 3.
3. Improvement of the Discussion and Introduction sections to improve overall structure, flow of arguments, and take-home message.

Responses to Reviewer #1:

Note: In the response text (blue, italic), line numbering refers to the cleared version of the revised manuscript (V4).

The authors have significantly revised the paper and put considerable effort into addressing my concerns. I appreciate this. The new analyses, in which new multifunctionality metrics are presented have given greater insight and relevance to the results (but additional caveats should be added to the paper surrounding their presentation and interpretation). However, this was a frustrating paper to re-review as many aspects of the paper are not fully worked up or presented to a sufficient standard. In general, several sections need re-writing, both at a within sentence level but also in terms of overall structure and content, to improve the logical flow of arguments and the presentation of ideas. These are detailed below but in particular the introduction, the ecological multifunctionality part of the results and the mid-discussion need attention.

Response: We appreciate the help to improve our manuscript. Based on this comment, we revised the manuscript thoroughly to improve the logical flow of arguments and the presentation of ideas.

While the new analysis is improved and does provide more insight, it is not properly justified or explained, or properly compared to the economic approach. Also, the landscape simulations are a nice addition but not clearly presented or interpreted. As the data is taken from 1 experiment/field, it's perhaps a bit much to extrapolate these findings to landscape scales. The

site in question will show only a small range of possible values for the different land use types, and so this part needs to be treated with caution. I actually feel it might be easier to omit it altogether.

Response: We appreciate this advice to improve our manuscript. Based on this comment (and on advice of reviewer 3), we omitted the landscape simulation approach.

I also have some concerns about how the Peter data was used. The services surveyed there were 'final benefits' (sensu cascade model), but most of what is measured here is regulating/supporting level (depending on which ES classification scheme is used). The authors reallocate or guess at stakeholder priorities to the point I feel this data can only be said to loosely resemble the original survey, and the measures to represent what stakeholders said they prioritised. Wording should be toned down and caveats added to acknowledge this.

Response: To address this criticism, we more clearly discuss the approach and acknowledge potential caveats at multiple parts of the revised manuscript (see specifically, lines 881-887).

A small point, but I also suggest calling the measures ecological or stakeholder based and or monetary or economic multifunctionality, and writing these in full. It would take up a few words but make the paper much easier to read- I constantly had to keep reminding myself which MF measure I was reading about.

Response: To address this point, we now always refer to ecological multifunctionality and economic multifunctionality, and we do not abbreviate these terms.

Below are more detailed points which I hope can be useful in revising the paper.

Specific points (Line numbering refers to the cleared version of the manuscript V3)

Throughout- replace 'farmland' with cropland or arable land (ackerland), as farmland often means any agricultural land including that used for livestock.

Response: Agreed. We changed farmland to cropland throughout the revised version of the manuscript.

25-28 intro ok here, but there isn't really a 'research gap' statement.

Response: Agreed. We added a respective text (line 27).

26. Omit functions or revise to account for these as underpinning services

Response: Changed (see line 26-27).

36. not clear what ecological multifunctionality is here- one of the stakeholder weighted measures or all of them?

Response: To increase precision, we changed "multiple weighting scenarios" to "three out of four weighting scenarios based on stakeholders' preferences" (see line 38-39).

37 according > based upon...does intensively....show higher...when compared to

Response: Changed.

43-45- conclusion still feels a bit generic and not really closely tied to results. E.g. you now know that the current prevailing land management set up suits the farmer (which is why they manage as they do) but not the rest of society, so it indicates that the rest of society needs to incentivize/pay the farmer to get the outcome they want/need for sustainability.

Response: We adapted the Abstract's conclusion accordingly.

58. this point was better covered by Byrnes et al 2014 than Manning 2018.

Response: Changed.

58-60. more that this approach is more meaningful if working at the level of ecosystem services

Response: Agreed. We changed the text accordingly (see lines 65-67).

64- would be good if the intro before getting here stated that:

1. we need ways of measuring overall social impact of global change on overall performance of ecosystems and that
2. Economic valuation and multifunctionality measures are a way of doing this but there are pros and cons to different approaches...

Response: Changed (see lines 63-64, and 64 following).

68- allows consideration of other aspects such

Response: Changed (see lines 79-80).

70 – trade-offs between stakeholders a key aspect here: see also Linders et al 2021 People and Nature.

Response: Changed and reference added (see lines 80-84).

72- 'can be used to approximate'- more accurately 'can be used as indicators'.

Response: Changed (see lines 128).

Stakeholders>Stakeholder groups

Response: Changed throughout the manuscript.

80- see Le Provost et al 2023, NEE for a more up to date reference on land-use-biodiversity-EMF relations (Hector and Bagchi did not look in an agricultural context from what I remember)

Response: We added the reference of Le Provost et al. (2023) and removed the reference of Hector and Bagchi (2007).

80 Here > However, at the level of ecosystem services (and move sentence 82-84 before this one)

Response: Changed (see line 91).

90- found>received

Response: Changed.

90-94. Poorly written- please revise

Response: We revised this section of the manuscript to improve readability (see lines 96 following).

100- see also <https://www.nature.com/articles/ncomms9159> and the many papers by Fernando Maestre and others looking at climate MF relationships

Response: Agreed. We added the publication of Jing et al. (2015) and of Maestre et al. (2012) (see line 112).

102- Fischer et al., 2010 does not look at multifunctionality from what I can recall

Response: Agreed. We removed the reference to Fischer et al. (2010).

107-111 – the gap is not expressed clearly here but it needs to be. There are lots of papers on how changes to climate and land use will affect economic properties – IPCC and IPBES reports are full of them, so do you mean how these drivers affect EMF measures that include economic properties? Neyret et al 2023 had some economic properties mixed with ecological ones, but done coarsely, and I can think of another Grass et al, that trades off economic and ecological multifunctionality, Grass et al 2020 <https://www.nature.com/articles/s41467-020-15013-5> but there is something different here to those in these papers- please clearly highlight the novelty and need for the approach used.

Response: Thank you for helping us sharpening the novelty statements in our manuscript. We rewrote the corresponding paragraph to highlight the lack of studies that examine the combined effect of different land-use types and future climate on combined ecological-economic multifunctionality measures, specifically in the context of agriculture and based on field data. While we mention the study of Neyret et al. (2023) at different parts of the manuscript, the study of Grass et al. (2020) that contrasts ecological multifunctionality of different land-use types with economic benefits for farmers (corresponding to the 'economic multifunctionality for farmers' in the manuscript presented here) does not provide any indication for the economic value of all ecosystem services considered. We further highlighted the novelty of expressing all ecosystem service levels in monetary terms. We added another paragraph highlighting the relevance of the approach used in this manuscript that allows us to provide a more comprehensive understanding of the complex interplay between land-use and climate change, and ecosystem multifunctionality in agricultural ecosystems, ultimately guiding informed decision-making for sustainable land management (see lines 119-128).

137-139 As much as I'd like them to, I'm not sure if people really appreciate soil biodiversity- most don't even know it exists! It's only just beginning to be appreciated by professional ecologists, though interest is rising in the popular imagination too, which is great. I would imagine a survey of preferences/valuation of biodiversity would put this group very low. In the figure it seems its classified as supporting anyway (Table 7 confirms this). Still, we could probably debate weightings, ES classification schemes etc. eternally. While I don't fully agree with all the choices made in this aspect, I accept them as they are justified at least. But the

divergence of what Peter et al measured and what is presented here needs to be acknowledged (see later comments).

Response: We added a section discussing the caveats in the transfer of stakeholders' preferences from Peter et al. (2021) to the different set of ecosystem services within our study (see lines 881-887).

Figure 1. Inessential, but pollution is a human value, so should maybe be in services- a flow of: nitrogen surplus, soil nutrient concentration (as the indicator) and water quality as the service would be more accurate there. Flower production>flower cover may also be more accurate.

Response: We changed the figure and the wording throughout the text:

- 'water quality' instead of 'water regulation'
- 'Soil nutrient concentration' instead of 'Water pollution'
- 'Flower abundance' instead of 'Flower cover' (for the ecosystem function)

164- stakeholder groups

Response: Changed throughout the manuscript.

164-165- would be useful to state roughly what the main differences in preferences were here, so results can be understood, and that this data was manipulated considerably.

Response: We added a sentence providing an overview of the main differences in preferences (see lines 186-189).

190- the response letter suggested some landscape simulation was done but I don't see this described in the approach part? (Reading on this either has to be more focal or omitted)

Response: We omitted this part, as proposed also by reviewer 3.

191-211- the results are somewhat listed off here- could do with some explanation or highlighting of major trends and patterns

Response: We added explanations and highlighted major trends and patterns within the text.

214- this simulated landscapes part comes in out of the blue- it needs to be in the approach section. The results also need to be presented in a more structured way so that they are more distinct and the messages behind each part are clear. (it might be easier to omit this part)

Response:

- *We understand the criticism and thus omitted this part in the revised version of our manuscript.*

Figure 2. Stakeholder weightings or contribution of individual ES not really shown anywhere so hard to understand what is behind these trends

Response: We added a sentence in the Introduction explaining main trends in stakeholder weighting (see lines 186-189) and added those main trends to the figures caption (see lines 255-258).

Table 2. A lot of abbreviations presented- what are all these? Please write out in full. The values are percentages presumably? Better presented as 4 pie charts perhaps?

Response: We omitted this part.

Table 3- a main paper item? Perhaps present key values in text and put rest in appendix.

Response: We shifted the respective part to the Supplementary material and changed table numbering throughout the manuscript.

257- these results are also listed off, but should be compared to the other EMF metrics. What changes when different services are introduced or omitted- make it clear and easy for the reader. The whole economic section is written in a different style and has a different feel to the section preceding it. Please harmonise.

Response: We revised the respective section according to this comment and put into focus the comparison of the economic and ecological approach.

268 not not

Response: Corrected.

304- please give a quantify this strong correlation – is it in the realm of 0.4 or 0.99- I've seen both described as strong.

Response: We added the correlation coefficient range (see line 308 following).

306- this difference sounds interesting- can you expand and say why its different

Response: Agreed. We expanded the sentence accordingly (see lines 310 following).

Figure 3. Really nice figure.

Response: Thank you!

Table 4. see comments on Table 3

Response: We shifted the respective part to the Supplementary material and changed table numbering throughout the manuscript.

336. This section feels a bit of an afterthought, a correlation is seen but would you really expect soil fauna diversity to be driving all these ES? A correlation is found but not much is done to properly separate the effects of diversity from other drivers (all the low soil BD are also arable fields so a totally different system). Options: expand, omit, place strong caveats, or justify more strongly.

Response: We understand the reviewer's criticism and addressed this point in two ways. First, we present evidence from the literature that soil biodiversity is a main driver of ecosystem multifunctionality across land-use types (Wagg et al. 2014, Soliveres et al. 2016, Schuldt et al. 2018, Delgado-Baquerizo et al. 2020). Second, we now clearly state that we can only show a correlation here that does not prove causality, and that future experimental work is needed to

verify causality (see lines 534-538). As a positive example, we cite Delgado-Baquerizo et al. (2020) again who presented a nice combination of observational and experimental results.

361- but a clear comparison was not made in the results

Response: We added a clear comparison in the results section (1st paragraph of section 3.2).

365- please clarify what the differences were

Response: we added a sentence highlighting the main differences between the economic and ecological (based on stakeholders' preferences) multifunctionality measures (see lines 373-375):

- *Land-use type: Effect on both the economic and the ecological multifunctionality measure*
- *Future climate: no effect on economic multifunctionality, negative effect on ecological multifunctionality.*

366- this study or any study?

Response: We clarified to express that any assessment of ecosystem multifunctionality will always have to choose a subset of ecosystem functions or services based on subjective perception of ecosystem service relevance and on data availability (see line 375 following).

369- representative of what?

Response: We expanded the sentence (see line 379 following).

387- discussion seems quite climate focused, relative to its prevalence in the main paper

Response: An important aspect of the study is ecosystem multifunctionality in different land-use types as affected by a projected future climate. When revising the manuscript, we paid attention to a balanced representation of findings that also highlights the novelty of our study.

397-398- was the data to support this argument presented?

Response: Yes, the data that supports this argument is presented in Figure 6 (overview of the performance of the individual ecosystem functions), where it shows that only 3 functions (yield, flower cover, aboveground decomposition rate) are negatively affected by future climate (and, by that, must be the drivers of overall (negative) effect of future climate ecosystem multifunctionality). We added a sentence to highlight this effect in the Results (see lines 231-233 and 245-247) and added an explanation in the Discussion (see lines 409-417).

407-409 Note that the survey of of Peters et al asked stakeholder to weight services at the final benefits level only, and so regulating/supporting services underpinning final benefits are not included. This will likely have influenced your results, which include a lot of these regulating services. This needs to be discussed.

Response: We agree with the reviewer and appreciate this helpful comment. Accordingly, we added a paragraph that discusses the varying classifications of services into different service categories and the implications for the results (see line 878 following).

404-419 I found this passage a bit confusing at it seemed to start with a discussion of ecological EMF, then discuss economic- needs revision/clarification.

Response: We rewrote and clarified the respective passage.

458. Unclear sentence

Response: We clarified the respective sentence (see lines 486-487).

483- farmland might be better called arable cropland here and elsewhere as pasture is also farmland

Response: We changed the term 'farmland' to 'cropland' throughout the manuscript.

497- new paragraph here

Response: We added a new paragraph.

517- certain threshold of what?

Response: We clarified that we refer to a certain threshold of stress level, which will depend on the respective ecosystem (see line 548).

526- the landscape composition part is very underdeveloped- expand or omit. The data appears rather shoehorned into this approach and I'm not sure if its really suitable for it, given that its effectively taken from a single field, and represents only a small subset of landscape services.

Response: Agreed. We omitted this part.

529- not clear what is meant by linear models here

Response: Linear models in the context of landscape multifunctionality refer to models that do not take into account feedback loops when landscape composition changes at large spatial scale (e.g., while a linear model would suggest to increase the share of land allocated to organic farming to maximise multifunctionality, it would neglect that this could lead to decreases in prices for organically-produced crops (and by that, to decreases in overall multifunctionality)). Still, as we decided to omit this part anyway, we did not add any further explanation.

539 indifferent- do you mean moderate/intermediate or similar? I think the latter

Response: True, we meant similar (but omitted this part anyway).

548 onwards- discussion is much stronger here

Response: Thank you.

651- how well does microbial biomass approximate water- there are a lot of other factors influencing microbial biomass so maybe not that great?! (also plant growth, I thought we were talking about microbes?)

Response: We clarified the sentence accordingly and noted the dependency of microbial biomass on other factors such as temperature, soil pH, organic carbon availability, and oxygen level (see lines 672-674).

826- mean

Response: We changed “arithmetic mean” to “mean” (see line 849).

832- biodiversity conservation in Peter et al was not a supporting service so this is questionable, also values have very much been assumed here, and others are missing. Acknowledge and discuss

Response: Agreed. We clarified the differing classification of the service biodiversity in the study of Peter et al. (2021) and the presented study (see lines 881-887).

844- where do these numbers come from? Peter et al show farmers giving around 15% of their score to biodiversity (and then as a cultural service), much lower than the 29% given here. I guess this is 29% of the parts that there was data for? Given the amount of data manipulation to fit the services measures it might be best to say these weightings are loosely based or inspired by this data as they are quite far removed from them.

Response:

- *We added an overview of the process of preference transfer to the supporting information (see Table S15).*
- *We further added an explanation to clarify that the weightings employed in this study are only partly based on the original data due to inherent differences in data availability (see lines 881-887).*

856- Neyret et al used aggregated measures of value from land uses when combining land classes – how was this done here?

Response: We omitted this part.

Table 7. Peter classified biodiversity as a cultural service (enjoyment of wildlife etc).

Response: Thank you. We clarified the deviation of ecosystem service classification in the table’s caption and throughout the text.

Table 7. Should the two food scores not be summed rather than averaged as they when combined represent the overall importance of food.

Response: Considering that within the experimental setup of the Global Change Experimental Facility, where we obtained the data for this study from, some land-use types providing only livestock production, some only food production, we find that summing the two food scores would overestimate the relative importance of the ecosystem service ‘food production’ as used in this study. We now clarified this in the text (see lines 862 following).

1540 Notes need to be deleted.

Response: Deleted.

Table 16 Is poorly formatted and can’t be read

Response: We changed the formatting.

Responses to Reviewer #2

The authors have accommodated all my comments to my satisfaction. Very interesting and important contribution. Well done.

Response: We appreciate this feedback and thank the reviewer again for the very helpful comments on the earlier draft of our manuscript.

Responses to Reviewer #3:

Thank you for the opportunity to review this new version of the manuscript. The authors have revised significant portions of the text to improve clarity regarding the protocol, analyses, and multiple metrics used in the study. They integrated new data of “real” ES prioritization from an earlier study, and conducted a whole new set of landscape simulation analyses to identify landscape compositions that maximize their different indices of multifunctionality. I commend them for these in-depth revisions. Unfortunately, one major issue of the previous version of the manuscript (also noted by Reviewer 1) was the lack of a tighter framing and take-home message of the study. This has been addressed but only partially (l. 60+). Further, the new analyses are not introduced at all and come “out of the blue” in the result section, which is a shame considering the effort I expect went into them. I do think the findings, and new analyses, are of interest but I strongly suggest additional effort, in particular in the introduction, to justify their scientific relevance and highlight research gaps as well as objectives and specific research questions of the study. This would also help highlight a few main take-home messages in the discussion. Please also find some more specific comments below.

Response: We appreciate the help to improve our manuscript. Based on this comment, we revised the manuscript thoroughly to improve the logical flow of arguments, the presentation of ideas and the take-home message and omitted the landscape-multifunctionality analyses.

Specific comments (line numbering refers to markup version of manuscript V3)

L. 39: be more specific: which land-use most mitigates climate change?

Response: In general, sustainably-managed cropland is most strongly affected by future climate. This is now more clearly phrased in the revised version of our manuscript.

New analyses regarding optimal landscape composition are not mentioned in the abstract

Response: We omitted this part as recommended by reviewers 1 and 3.

l. 65 onwards: I appreciate the justification of both indices of multifunctionality. Part of this could go later in the introduction, after some hypotheses are more clearly introduced, and along with some research questions (see below for l. 115-116). This section is mostly clear about what ecological multifunctionality brings compared to the economical one, but does not say why the economical one is needed.

Response: We added the justification for the economic approach of measuring ecosystem multifunctionality, that allows an assessment of the benefits for society at large with a unified, monetary metric (see lines 75 following) and restructured the Introduction.

One additional argument to this part could be that traditional (non-weighted) approaches of multifunctionality do not take into account the trade-offs between the priorities/demands of different people, and thus the possibly contrasted benefits they gain from ecosystems – which both the egEMF and to some extent the enEMF (with the different benefits for farmers and overall) can integrate.

Response: Agreed, thank you. We added the additional argument (see lines 80-84).

L. 67: I don't understand what "driven by EMF" means here

Response: True, this is redundant → removed.

I. 71 (also 104): contrary -> conversely

Response: Changed.

I. 74: the equity part comes out of the blue here, it would need some context regarding why equity is considered or what it brings to the topic. I would suggest to just skip equity and focus on multifunctionality (also in the landscape optimization: use only multifunctionality as criteria) or to add a full introduction and discussion of equity in terms of trade-offs between different stakeholders, limiting conflicts, etc + relevant literature (but this would probably lead to too many concepts introduced in this already dense paper, the safest might be to skip it).

Response: We omitted this part.

L. 77 Definition box -> glossary

Response: Agreed. We changed the name of the box to Glossary.

I. 82-107: here the text jumps a bit from topic to topic climate – biodiversity – BEF – effect of climate on biodiversity – maybe could be streamlined a bit to better introduce the next paragraph? A bit of reorganization and a few additional words would help to make the arguments (which are already there but more or less implicitly) more explicit (e.g. in the line of "we know climate will affect individual services, but because people value services differently the overall impact on stakeholders well-being and economic benefits remain unknown"; "climate change (combined with anthropogenic impact) affects biodiversity, but as BEF relationships might change too the overall impact on ecosystem functioning is unclear"

Response: We agree with the reviewer and rewrote the Introduction accordingly.

I. 115-118: this part should be expanded (maybe as a new paragraph) to justify and introduce all the analyses covered in the paper, including the BEF and landscape simulation analyses. The gap addressed by this study also goes beyond the use of monetary units as this is one among multiple indices used to address the combined effects of land use and climate change on these systems

Response: We agree with the reviewer and expanded the respective text passage.

I. 147: unclear – what is not included is the intrinsic value? But the insurance value is, is that right?

Response: Correct. We clarified this in the revised version (see lines 153-166).

I. 168 onwards: some details (eg. Data exclusion for the first two years, process to average indicators across years, detailed economic quantification, sensitivity analyses) don't need to be repeated here from the methods/results

Response: We agree with the reviewer and removed the respective text passages.

I. 203: the broader, expected outcome of the paper is currently a bit hidden here following the description of sensitivity analyses – could be moved to elsewhere in the intro or to the discussion

Response: While revising and streamlining the manuscript, we paid particular attention to highlight the main outcomes. One of the main findings is the positive relationship between ecosystem multifunctionality and soil biodiversity, because it is in line with previous studies in other ecosystems and underlines the need of land managers to conserve soil biodiversity.

I.234 onwards: very long sentences, difficult to follow.

“The weighting according to preferences of the tourism...” and similar could be shortened to e.g. tourists egEMF here and elsewhere.

Response: We reduced sentence length and shortened terms (“ecological multifunctionality under weighting according to preferences of farmers” → “for farmers, ecological multifunctionality...”).

I. 252: I suggest removing the equity-related analyses (or fully introduce them in the introduction).

Response: Agreed - we omitted this part.

I. 252: long and complex sentence, rewrite?

Response: We omitted this part as recommended by reviewer 1 and 3.

Fig 2: I might have missed it in the results and discussion, but it seems that the OF treatment, despite having relatively higher EMF than the other land uses, is clearly the one that would have the largest decrease under climate change; Does that mean that OF is actually more sensitive to climate change, or simply has more “room” to decrease as it starts from a high value? Maybe worth discussing

Response: Very important point - thank you. In fact, the reviewer is right that organic farming showed the most pronounced absolute reduction in ecosystem multifunctionality under the future climate. At the same time, all other land-use types experienced reductions in ecosystem multifunctionality, and organic farming was still the land-use type that showed highest levels of ecosystem multifunctionality under the future climate, underlining its benefits under current and

future climate. This is now more clearly described in the revised version of the manuscript (see line 234 and 442).

I. 409 onwards: is it still the sensitivity analyses section or a new one?

Response: Clarified!

I. 496: In fig 5 the light blue pvalue (which I expect corresponds to the future climate) says 0.06, not < 0.01, homogenise (or explain the difference)

Response: Changed (typo).

Fig 5 what is the black line? Some confidence intervals should be added. Is the interaction biodiversity x climate significant? What is the R2 and pvalues in black on top?

Response:

- *The black line represents the effect of Multidiversity on ecosystem multifunctionality across all climate types (without differentiating ambient and future climate*
- *The R2 and p values in black refer to the regression across all climate types (shown as black line in the earlier version of the manuscript). We clarified this in the figures caption.*
- *Confidence intervals: added!*
- *The interaction biodiversity x climate is not significant (The coefficient for Multidiv:Climatefut is 0.17695, the standard error for this coefficient is 0.15679, the t-value is 1.129, The p-value associated with this coefficient is 0.2649) → In the context of our regression model, this means that there is not enough evidence to conclude that the effect of soil multi-diversity on ecosystem multifunctionality (without soil biodiversity) differs significantly between the two levels of the climate scenarios.). These results are now more clearly explained in the revised version of the manuscript.*

Discussion: I found the beginning and end of the discussion interesting and useful, the middle part is a bit harder to follow as it switches back and forth from repeating some results, discussing specific services (eg leaching), or study site specificities or broader land-use change considerations. I don't really have a constructive suggestion on how to improve this, unfortunately, but maybe having separate paragraphs for the different themes and focusing on explicitly answering/discussion the different research questions would help.

Response: In response to this comment, we paid particular attention to present a clear and streamlined version of the Discussion, but - in this case - we did not rewrite completely, to maintain the flow positively evaluated by the other two reviewers.

I. 1133: maybe I missed it, how do you calculate landscape-level service supply (e.g. average, sum? Do you average biodiversity values or use gamma diversity (it should be gamma diversity)?

Response: We omitted this part as recommended by reviewer 1 and 3.

I. 1141: technically power is not needed to calculate community-level multifunctionality, and you don't use it anyways, so you can remove this sentence

Response: We omitted this part as recommended by reviewer 1 and 3.

Reviewers' Comments:

Reviewer #1:

Remarks to the Author:

This version of the paper has addressed most of my previous concerns. However, some parts are still a little lacking in clarity and overall, the paper could probably be improved a bit in terms of the clarity and precision of writing, and presentation. It's also still a little bit difficult to see to a clear objective and take-home message, but I do think the paper is greatly improved in this respect. I have a few suggestions and queries remaining, none of which are major. Hopefully these can help improve the paper a little more.

General points:

Curious here, and this might make a discussion point, but how different is economic multifunctionality to total economic value (TEV), as presented in much earlier ecosystem service valuation work? They seem very similar to me- or is there a key difference?

Correlations between the different multifunctionality scores are presented but there is not much in the way of analysis as to what differences there are and why- adding a little more on this would help highlight the novel aspects of the paper.

The soil biodiversity result feels somewhat oversold- it is correlative and not so strong, with other covarying factors not accounted for, and strongly confounded with management (see later specific comments for more). I would suggest toning the wording down a bit here.

It might be nice to say what the different stakeholders prioritise, or to present the weightings, (not essential but could be in the first figure?) to give transparency to the results. In general, it should be clearer in the main text that these are very loosely based on the Peter et al. data, as some ES even move broad categories (cultural to supporting). The text in the methods is clear here, but it needs to be in the main paper too.

Discussion is a bit rambling and could be more concise and synthetic – see some specific comments on this.

Acknowledging that multifunctionality is demanded and delivered at a higher scale would also be good – i.e. from a mix of these land use types. While the analysis on this part was removed, and this is probably for the best, it should probably be noted that land use types interact spatially to determine multifunctionality (Le Provost et al 2023, NEE) and that at landscapes scales different land types provide different services (Neyret et al 2021, cited, and 2023, Nat Sust), so it's not just about covering whole landscapes with a single 'best' land use. The current discussion implicitly leans a bit towards this, so I feel it's worth adding a small caveat here.

Specific comments

29- representing > underlying

63 - allows to assess > allows for assessment of

65- typically based (there are a range of approaches)

83-84- is more that it allows for assessment of how well an ecosystem meets the needs of different stakeholders

97 of > in

126- which may ultimately guide

Glossary; should ecosystem function include physical? Biophysical as purely physical processes- like wind or erosion- would not be an ecosystem function

Figure 1. A few of these are misclassified or labelled in my view. I suggest:

Yield should perhaps not be a function as it's only via the human lens of agricultural production that decides which part is the used element. Seed production for crops /aboveground biomass production for fodder instead?

sequestration > storage

Soil nutrient concentration > nutrient leaching rate

200- Allan et al 2015 did not estimate an effect of soil biodiversity so this should be removed

216- typically indicates are more general result than that of this study
228- not essential but locals may be more clearly labelled as local residents
373- this seems a bit strong- I would say quite a lot of services are missing here and it is worth saying that the outcomes could differ if they were included.
392- the references given refer to biodiversity-stability relationships rather than land use types, although some are more diverse than others – revise wording or references
392 and 396- towards>to
404-408- a more general point about the capacity of this single measure to represent cultural ES may be more relevant
417- take home message/summary point seems to be missing from this paragraph
418- all MF types/metrics?
436 .This demonstrates
443-333- sentence needs clarifying
444-452- this is well established information and could be more concise
475- delete also
500- still find this C value surprisingly high! See later notes on methods.
500-517- links between sentences and points here could be clearer, and the 513-517 part is rather vague and hard to derive meaning from
526- might be good to say what else the benefit could come from here- e.g. that high SOM from lower tillage will lead to both high soil BD and other function increases like water retention and nutrient cycling. Can't be dealt with here beyond an acknowledgement of these interrelations but disentangling the cause and effect of SOM, soil BD and other functions would be a very worthwhile endeavour to clear this up! Cross SOM content with soil bd experimentally?
541-544- seems a strange point to make given the result, which goes the other way, does it not?
544- research of what sort- statement is a bit vague
546-550- show rather than tell here- i.e. say what additional information is provided by the combined approach.
553-555- how and where was it used? Vague. Do you mean it could be used?
585 value to
614 number of replicates per treatment combination should be presented
769- would be good to see some referencing/justification of why these measures represent soil health.
822- equations were not displaying in the version of this I downloaded/reviewed
838- relevant for this study- is a bit vague- relevant in what way?
892- a quick google indicates this carbon price is over double current market prices, and that even the alternate 'low end' cost of sensitivity analysis of 90 is very high, based on the range of market values (40-90 USD).
931- equations not showing in this section either
Figure S1. Might be good to show the original data here so the difference can be seen- i.e where the points came from and where they were reallocated to
Figure S2, S3. Would be nice to highlight how these figures differ from the main ones in the caption, as it always gets a bit confusing when there are lots of sensitivity analyses
Supplementary tables- the column width should be edited to allow these to be read more easily- e.g. words wrapping round
Table S15- I welcome this table but find it hard to follow the procedure used- can it be clarified?

Reviewer #3:

Remarks to the Author:

Thank you for providing a revised version of this manuscript. Many of the points raised previously have been addressed.

Among the key points remaining, I would suggest to be clearer about the priority data – this is not “from survey data” as suggested in the abstract in a few other places, but only inspired by / adapted from previously published data in another region. This is explained in the methods but should be made clear early on. Besides, while the “flow” of the manuscript has overall improved, there are still many long or convoluted sentences (some of them noted below) which would benefit from additional clarification.

Detailed comments

l. 33 "i.e."?

l. 35: this was not really survey data, more inspired from previous study?

l. 40: "higher multifunctionality than sustainably-managed grassland"?

l. 43: "The economic multifunctionality measure was about 1.7 higher"

Intro

l. 105: "However, empirical work"... the sentence is a bit convoluted, maybe rephrase to "and a study that simulated... found no significant difference"

l. 115 very long sentence

l. 186 why "whereas" (this word is used in many places in the manuscript where I'm not sure it has the correct meaning)? + the priority data was not directly derived from survey data, but rather "inspired" or "adapted from" previously published stakeholder data.

Results

l. 340: "correlates significantly positive" -> "has a significant and positive correlation?"

Fig 2 caption: provisioning and cultural services

l. 291: Unimportant, but as the price scenarios for the main results haven't been introduced yet, providing the detailed prices might be unnecessary here (you could start with "the finding that...." And provide specific values in the methods only)

Discussion

l. 400: not sure "driver" is correct term – they are the components of multifunctionality that have the strongest response

l. 425: with twice

l.561: demand -> "need"?

Methods

l. 828: The first two first years

Report how we have revised the manuscript:

Note: Line numbering cross references refer to the markup version of the manuscript.

Sustainable land management enhances ecological and economic multifunctionality under ambient and future climate

For Nature Communications in response to the reviewers' comments

We are grateful for the constructive comments provided by the editor and the reviewers. We feel that revising the manuscript and addressing these comments has made it much more convincing and understandable to the broad readership of Nature Communications. In the following, we explain all changes we have made in response to the review reports. To this end, we reproduce the reviewer comments and provide the respective answers in *blue*.

Our main revisions have been the following:

1. Clarification of the origin of the stakeholder preferences (weightings) based on a transformation process of the survey data from Peter et al. (2021) and emphasizing the deviations from the original data that result from the transformation process.
2. Improvements of flow / readability.

Responses to Reviewer #1:

This version of the paper has addressed most of my previous concerns. However, some parts are still a little lacking in clarity and overall, the paper could probably be improved a bit in terms of the clarity and precision of writing, and presentation. It's also still a little bit difficult to see to a clear objective and take-home message, but I do think the paper is greatly improved in this respect. I have a few suggestions and queries remaining, none of which are major. Hopefully these can help improve the paper a little more.

Response: We appreciate the continued help to improve our manuscript. We also went through the manuscript again to improve clarity and precision of writing and presentation.

General points:

Curious here, and this might make a discussion point, but how different is economic multifunctionality to total economic value (TEV), as presented in much earlier ecosystem service valuation work? They seem very similar to me- or is there a key difference?

Response: While the two concepts are related, there is a key difference: economic multifunctionality only takes into account the current economic values related to ecosystem functions. The total economic value includes all values attached to the ecosystem, including existence and bequest values, which are not directly related to current ecosystem functions. We added a respective caveats statement (Introduction, lines 544 ff.).

Correlations between the different multifunctionality scores are presented but there is not much in the way of analysis as to what differences there are and why- adding a little more on this would help highlight the novel aspects of the paper.

Response: We agree and added a discussion of the main findings from the correlations between different multifunctionality measures (see Discussion, lines 673 ff.).

The soil biodiversity result feels somewhat oversold- it is correlative and not so strong, with other covarying factors not accounted for, and strongly confounded with management (see later specific comments for more). I would suggest toning the wording down a bit here.

Response: Agreed; we added a paragraph to tone down the wording (e.g., Abstract, line 47, Discussion, lines 615, ff.).

It might be nice to say what the different stakeholders prioritise, or to present the weightings, (not essential but could be in the first figure?) to give transparency to the results. In general, it should be clearer in the main text that these are very loosely based on the Peter et al. data, as some ES even move broad categories (cultural to supporting). The text in the methods is clear here, but it needs to be in the main paper too.

Response: Agreed. We changed the respective section in the Abstract (line 36), Introduction (lines 197, ff.), and in the Glossary (line 137) to make it clear that the weighting factors applied in this study are only partly based on the original survey data.

Discussion is a bit rambling and could be more concise and synthetic – see some specific comments on this.

Response: We improved flow and readability in the Discussion.

Acknowledging that multifunctionality is demanded and delivered at a higher scale would also be good – i.e. from a mix of these land use types. While the analysis on this part was removed, and this is probably for the best, it should probably be noted that land use types interact spatially to determine multifunctionality (Le Provost et al 2023, NEE) and that at landscapes scales different land types provide different services (Neyret et al 2021, cited, and 2023, Nat Sust), so it's not just about covering whole landscapes with a single 'best' land use. The current discussion implicitly leans a bit towards this, so I feel it's worth adding a small caveat here.

Response: Agreed and respective caveat added (lines 639, ff.).

Specific comments

29- representing > underlying

Response: Changed.

63 - allows to assess > allows for assessment of

Response: Changed.

65- typically based (there are a range of approaches)

Response: Changed.

83-84- is more that it allows for assessment of how well an ecosystem meets the needs of different stakeholders

Response: Adapted accordingly.

97 of>in

Response: Changed.

126- which may ultimately guide

Response: Changed.

Glossary; should ecosystem function include physical? Biophysical as purely physical processes- like wind or erosion- would not be an ecosystem function

Response: Adapted accordingly in both Glossary (line 137) and Introduction (line 56).

Figure 1. A few of these are misclassified or labelled in my view. I suggest:

Yield should perhaps not be a function as it's only via the human lens of agricultural production that decides which part is the used element. Seed production for crops /aboveground biomass production for fodder instead?

sequestration>storage

Soil nutrient concentration > nutrient leaching rate

Response: Agreed and changed throughout the manuscript.

200- Allan et al 2015 did not estimate an effect of soil biodiversity so this should be removed

Response: Removed.

216- typically indicates are more general result than that of this study

Response: Clarified.

228- not essential but locals may be more clearly labelled as local residents

Response: Changed throughout the manuscript.

373- this seems a bit strong- I would say quite a lot of services are missing here and it is worth saying that the outcomes could differ if they were included.

Response: Clarified (lines 398, ff.).

392- the references given refer to biodiversity-stability relationships rather than land use types, although some are more diverse than others – revise wording or references

Response: Changed.

392 and 396- towards>to

Response: Changed.

404-408- a more general point about the capacity of this single measure to represent cultural ES may be more relevant

Response: A more general point was added.

417- take home message/summary point seems to be missing from this paragraph

Response: Take-home message added (lines 455, ff.).

418- all MF types/metrics?

Response: Changed (yes, for both the ecological and the economic MF measure).

436 .This demonstrates

Response: Changed.

443-333- sentence needs clarifying

Response: Clarified.

444-452- this is well established information and could be more concise

Response: Changed accordingly.

475- delete also

Response: Sentence deleted and following sentence adapted.

500- still find this C value surprisingly high! See later notes on methods.

Response? We added a statement referring to the rather high rate of soil organic matter accumulation (lines 557, ff.).

500-517- links between sentences and points here could be clearer, and the 513-517 part is rather vague and hard to derive meaning from

Response: We adapted this paragraph.

526- might be good to say what else the benefit could come from here- e.g. that high soil organic matter from lower tillage will lead to both high soil biodiversity and other function increases like water retention and nutrient cycling. Can't be dealt with here beyond an acknowledgement of these interrelations but disentangling the cause and effect of soil organic matter, soil biodiversity and other functions would be a very worthwhile endeavour to clear this up! Cross soil organic matter content with soil biodiversity experimentally?

Response: We expanded this section (lines 592, ff.).

541-544- seems a strange point to make given the result, which goes the other way, does it not?

Response: We adapted the section accordingly (lines 614, ff.).

544- research of what sort- statement is a bit vague

Response: Clarified.

546-550- show rather than tell here- i.e. say what additional information is provided by the combined approach.

Response: We adapted this paragraph (lines 625, ff.).

553-555- how and where was it used? Vague. Do you mean it could be used?

Response: We adapted this paragraph (lines 638, ff.).

585 value to

Response: Changed.

614 number of replicates per treatment combination should be presented

Response: Added (5 replicates for each land-use type-climate combination).

769- would be good to see some referencing/justification of why these measures represent soil health.

Response: Reference and justification added (lines 874, ff.).

822- equations were not displaying in the version of this I downloaded/reviewed

Response: Changed.

838- relevant for this study- is a bit vague- relevant in what way?

Response: We clarified this part (lines 941, ff.).

892- a quick google indicates this carbon price is over double current market prices, and that even the alternate 'low end' cost of sensitivity analysis of 90 is very high, based on the range of market values (40-90 USD).

Response: The value that we used is not the market price, but the social cost associated with the emission of one ton of carbon dioxide, which is 195 € / t in 2022 according to the German Environment Agency. In the sensitivity analysis, the market value within the European Emission Trading Scheme is used (90 € / t in 2021), which is found to be the most reliable indicator of the market value, as this is the fixed price that must be paid by companies on the compliance market (contrary to the voluntary market where prices are fluctuating highly).

We clarified this section (lines 1025, ff.).

931- equations not showing in this section either

Response: Changed.

Figure S1. Might be good to show the original data here so the difference can be seen- i.e. where the points came from and where they were reallocated to

Response: While we did not find a meaningful way to include the original data in the figure, as the original data used a different subset of ecosystem services, we recognise the importance of providing clarity on the transformation process. In response, we revised Table S15 to improve the presentation of the original and reallocated data, offering a more comprehensive overview of the transformation process.

Figure S2, S3. Would be nice to highlight how these figures differ from the main ones in the caption, as it always gets a bit confusing when there are lots of sensitivity analyses

Response: We added a statement regarding the main difference of the alternative weighting scenarios / sensitivity analysis to the main scenario to the figures' captions (line 1538, ff.).

Supplementary tables- the column width should be edited to allow these to be read more easily- e.g. words wrapping round

Response: Changed, link to google sheet added for Table S13 (tables will be handed in to editing as .xlsx files).

Table S15- I welcome this table but find it hard to follow the procedure used- can it be clarified?

Response: Changed, colors added to clarify the procedure.

Reviewer #3 (Remarks to the Author):

Thank you for providing a revised version of this manuscript. Many of the points raised previously have been addressed.

Among the key points remaining, I would suggest to be clearer about the priority data – this is not “from survey data” as suggested in the abstract in a few other places, but only inspired by / adapted from previously published data in another region. This is explained in the methods but should be made clear early on. Besides, while the “flow” of the manuscript has overall improved, there are still many long or convoluted sentences (some of them noted below) which would benefit from additional clarification.

Response: We appreciate the help to improve our manuscript and implemented your comments accordingly. We also went through the manuscript again to improve clarity and precision of writing and presentation.

Detailed comments

I. 33 "i.e."?

Response: Changed (deleted).

I. 35: this was not really survey data, more inspired from previous study?

Response: Agreed. We changed the respective section in the Abstract (line 36), Introduction (lines 197, ff.), and in the Glossary (line 137) to make it clear that the weighting factors applied in this study are only partly based on the original survey data.

I. 40: "higher multifunctionality than sustainably-managed grassland"?

Response: Changed.

I. 43: "The economic multifunctionality measure was about 1.7 higher"

Response: Changed.

Intro

I. 105: “However, empirical work”... the sentence is a bit convoluted, maybe rephrase to “and a study that simulated... found no significant difference”

Response: Changed.

I. 115 very long sentence

Response: Changed (split in two).

I. 186 why “whereas” (this word is used in many places in the manuscript where I'm not sure it has the correct meaning)? + the priority data was not directly derived from survey data, but rather “inspired” or “adapted from” previously published stakeholder data.

Response: We clarified the adaptation process of the original survey data from Peter et al. (2021) throughout the manuscript (e.g., Table S15). Further, we changed the word "whereas" throughout the manuscript.

Results

I. 340: "correlates significantly positive" -> "has a significant and positive correlation?"

Response: Changed.

Fig 2 caption: provisioning and cultural services

Response: Changed.

I. 291: Unimportant, but as the price scenarios for the main results haven't been introduced yet, providing the detailed prices might be unnecessary here (you could start with "the finding that..." And provide specific values in the methods only)

Response: Changed accordingly.

Discussion

I. 400: not sure "driver" is correct term – they are the components of multifunctionality that have the strongest response

Response: Changed.

I. 425: with twice

Response: Changed.

I.561: demand -> "need"?

Response: Changed.

Methods

I. 828: The first two first years

Response: Changed (deleted).